# Robustly learning the Hamiltonian dynamics of a superconducting quantum processor

Dominik Hangleiter [1,2,3,7] ✉, Ingo Roth [3,4,7] ✉, Jonáš Fuksa [3], Jens Eisert [3,5] & Pedram Roushan [6]

Precise means of characterizing analog quantum simulators are key to developing quantum simulators capable of beyond-classical computations. Here, we precisely estimate the free Hamiltonian parameters of a superconducting-qubit analog quantum simulator from measured time-series data on up to 14 qubits. To achieve this, we develop a scalable Hamiltonian learning algorithm that is robust against state-preparation and measurement (SPAM) errors and yields tomographic information about those SPAM errors. The key subroutines are a novel super-resolution technique for frequency extraction from matrix time-series, tensorESPRIT, and constrained manifold optimization. Our learning results verify the Hamiltonian dynamics on a Sycamore processor up to sub-MHz accuracy, and allow us to construct a spatial implementation error map for a grid of 27 qubits. Our results constitute an accurate implementation of a dynamical quantum simulation that is precisely characterized using a new diagnostic toolkit for understanding, calibrating, and improving analog quantum processors.

Analog quantum simulators promise to shed light on fundamental questions of physics that have remained elusive to the standard methods of inference[1,2]. Recently, enormous progress in controlling individual quantum degrees of freedom has been made towards making this vision a reality[3–6]. While in digital quantum computers small errors can be corrected[7], it is intrinsically difficult to error-correct analog devices. Yet, the usefulness of analog quantum simulators as computational tools depends on the error of the implemented dynamics. Meeting this requirement hinges on devising characterization methods that not only yield a benchmark of the overall functioning of the device [e.g.,[8–10]], but more importantly provide diagnostic information about the sources of errors.

Developing characterization tools for analog quantum simulators requires hardware developments as well as theoretical analysis and method development. With the advent of highly controlled quantum systems, efficient methods for identifying certain Hamiltonian parameters from dynamical data have been devised for specific classes of Hamiltonians. Key ideas are the use of Fourier analysis[11–17] and tracking the dynamics of single excitations[18–23]. For general Hamiltonian models, specific algebraic structures of the Hamiltonian terms can be exploited[24,25]. Generalizing these ideas, a local Hamiltonian can be learned from a single eigenstate or its steady state[26–31] or using quantum-quenches[32,33], an approach dubbed "correlation matrix method"[34]. Alternatively, one can apply general-purpose machine-learning methods[35–39]. More recently, optimal theoretical guarantees have been derived for Hamiltonian learning schemes[40–42] based on Pauli noise tomography[43,44]. Crucially, these protocols assume perfect mid-circuit quenches, which—as we find here—can be a limiting assumption in practice.

This recent rapid theoretical development is not quite matched by concomitant experimental efforts. The effectiveness of some of these methods has been demonstrated for the estimation of a small number

[1]Joint Center for Quantum Information and Computer Science (QuICS), University of Maryland and NIST, College Park, MD, USA. [2]Joint Quantum Institute (JQI), University of Maryland and NIST, College Park, MD, USA. [3]Dahlem Center for Complex Quantum Systems, Freie Universität Berlin, Berlin, Germany. [4]Quantum Research Center, Technology Innovation Institute (TII), Abu Dhabi, United Arab Emirates. [5]Helmholtz-Zentrum Berlin für Materialien und Energie, Berlin, Germany. [6]Google Quantum AI, Mountain View, CA, USA. [7]These authors contributed equally: Dominik Hangleiter, Ingo Roth. ✉e-mail: mail@dhangleiter.eu; ingo.roth@tii.ae

of coupling parameters of fixed two- and three-qubit Hamiltonians in *nuclear magnetic resonance* (NMR) experiments[45–48]. While in NMR, the dominant noise process is decoherence, in tunable quantum simulators such as superconducting qubits, trapped ions or cold atoms in optical lattices, *state preparation and measurement* (SPAM) errors, as we also demonstrate here, play a central role. Initial steps at characterizing such errors as well as the dissipative Lindblad dynamics for up to two qubits in a superconducting qubit platform have been taken recently[49,50]. Hamiltonian learning of thermal states has recently also been applied in many-body experiments as a means to characterize the entanglement of up to 20-qubit subsystems whose reduced states are parameterized by the so-called *entanglement Hamiltonian*[51–53]. The challenge remains to develop and experimentally demonstrate the feasibility of scalable methods for a robust and precise identification of Hamiltonian dynamics of intermediate-size systems subject to both incoherent noise and systematic SPAM errors.

In this work, we develop bespoke protocols to robustly and precisely identify the full Hamiltonian of a large-scale bosonic system and implement those protocols on superconducting quantum processors. Given the complexity of the learning task, we focus on identifying the non-interacting part of a potentially interacting system. We are able to estimate the corresponding Hamiltonian parameters as well as SPAM errors pertaining to all individual components of the superconducting chip for up to 14-mode Hamiltonians tuned across a broad parameter regime, in contrast to previous experiments. Given the identified Hamiltonians, we quantify their implementation error. We demonstrate and verify that a targeted intermediate-size Hamiltonian is implemented on a large region of the superconducting processor with sub-MHz accuracy in a broad parameter range.

To this end, building on previous ideas for Hamiltonian identification[19,24], we devise a simple and robust algorithm that exploits the structure of the system at hand. For the identification we make use of quadratically many experimental time-series tracking excitations via expectation values of canonical coordinates. Our structure-enforcing algorithm isolates two core tasks that need to be solved in Hamiltonian identification after suitable pre-processing of the data: frequency extraction and eigenspace reconstruction.

To solve the first task in a robust and structure-specific way, we develop a novel algorithm coined *tensorESPRIT*, which utilizes ideas from super-resolving, denoised Fourier analysis[54–56] and tensor networks to extract frequencies from a matrix time-series. For the second task we use constrained manifold optimization over the orthogonal group[57]. Crucially, by explicitly exploiting all structure constraints of the identification problem, our method allows us to distinguish and obtain tomographic information about state-preparation and measurement errors. In the quench-based experiment this information renders identification and verification of the dynamics experimentally feasible in the first place. We further support our method development with numerical simulations of different noise effects and benchmark against more direct algorithmic approaches. We find that in contrast to other approaches our method is scalable to larger system sizes out of the reach of our current experimental efforts.

Our work constitutes a detailed case study that lays bare and provides solutions for the difficulties of practical Hamiltonian learning in a seemingly simple system. It thus provides a blueprint and paves the way for devising practical model-specific identification algorithms both for the interaction parameters of bosonic or fermionic systems and more complex settings.

## Results
### Setup
We characterize the Hamiltonian governing analog dynamics of Google Sycamore chips which consist of a two-dimensional array of nearest-neighbor coupled superconducting qubits. Each physical qubit is a non-linear oscillator with bosonic excitations (microwave photons)[58]. Using the rotating-wave approximation the dynamics governing the excitations of the qubits in the rotating frame can be well described by the Bose-Hubbard Hamiltonian[59]

$$H_{BH} = \sum_i \left( \mu_i a_i^\dagger a_i + \eta_i a_i^\dagger a_i^\dagger a_i a_i \right) - \sum_{i \neq j} J_{i,j} a_i^\dagger a_j, \quad (1)$$

where $a_i^\dagger$ and $a_i$ denote bosonic creation and annihilation operators at site $i$, respectively, $\mu \in \mathbb{R}^N$ are the on-site potentials, $J \in \mathbb{R}^{N \times N}$ are the hopping rates between nearest neighbor qubits, and $\eta \in \mathbb{R}^N$ are the strength of on-site interactions. The qubit frequency, the nearest-neighbor coupling between them, and the non-linearity (anharmonicity) set $\mu$, $J$, and $\eta$. We are able to tune $\mu$ and $J$ on nanosecond timescales, while $\eta$ is fixed for a given setting of $\mu$ and $J$. Hence, the Sycamore chip can be used to implement time evolution under Hamiltonians of the form (1) at various parameter settings and is therefore an analog simulator. In a practical application such as in Ref. 60, it is crucial to benchmark how accurately the implemented dynamics is described by a targeted Hamiltonian.

Here, we focus on the specific task of identifying the values of $\mu_i$ and $J_{i,j}$. The corresponding non-interacting part of the Hamiltonian acting on $N$ modes can be conveniently parametrized as

$$H(h) = -\sum_{i,j=1}^N h_{i,j} a_i^\dagger a_j \quad (2)$$

with an $N \times N$ real symmetric parameter matrix $h$ with entries $h_{i,j}$, which is composed of the on-site chemical potentials $\mu_i$ on its diagonal and the hopping energies $J_{i,j}$ for $i \neq j$. The identification of the non-interacting part $H(h)$ of $H_{BH}$ can be viewed as a first step in a hierarchical procedure for characterizing dynamical quantum simulations with tunable interactions and numbers of particles.

The non-interacting part $H(h)$ of the Hamiltonian $H_{BH}$ can be inferred when initially preparing a state where only a single qubit is excited with a single photon. For initial states with a single excitation, the interaction term vanishes, hence effectively $\eta = 0$. Consequently, only the two lowest energy levels of the non-linear oscillators enter the dynamics. Therefore, referring to them as qubits (two-level systems) is precise. Specifically, we identify the parameters $h_{i,j}$ from dynamical data of the following form. We initialize the system in $|\psi_n\rangle := (\mathbb{1} + a_n^\dagger)|0\rangle^{\otimes N}/\sqrt{2}$ and measure the canonical coordinates $x_m = (a_m + a_m^\dagger)/2$ and $p_m = (a_m - a_m^\dagger)/(2i)$ for all combinations of $m, n = 1, …, N$. In terms of the qubit architecture, this amounts to local Pauli-$X$ and Pauli-$Y$ basis measurements, respectively. We combine the statistical averages over multiple measurements to obtain an empirical estimator for $\langle a_m(t)\rangle_{\psi_n} = \langle x_m(t)\rangle_{\psi_n} + i \langle p_m(t)\rangle_{\psi_n}$. For particle-number preserving dynamics, this data is of the form

$$\langle a_m(t)\rangle_{\psi_n} = \frac{1}{2} \exp(-ith)_{m,n}. \quad (3)$$

It therefore directly provides estimates of the entries of the time-evolution unitary at time $t$ in the single-particle sector of the bosonic Fock space.

In Fig. 1, we show an overview of the experimental procedure, and the different steps of the Hamiltonian identification algorithm. Every experiment uses a few coupled qubits, from the larger array of qubits on the device (Fig. 1a). On those qubits, the goal is to implement the time-evolution with targeted Hamiltonian parameters $h_0$, which are subject to connectivity constraints imposed by the couplings of the qubits. To achieve this, we perform the following pulse sequence to collect dynamical data of the form (3). Before the start of the sequence, the qubits are at frequencies (of the $|0\rangle$ to $|1\rangle$ transition) that could be a few hundred MHz apart from each other. In the beginning, all qubits are in their ground state $|0\rangle$. To prepare the initial state, a $\pi/2$-pulse is

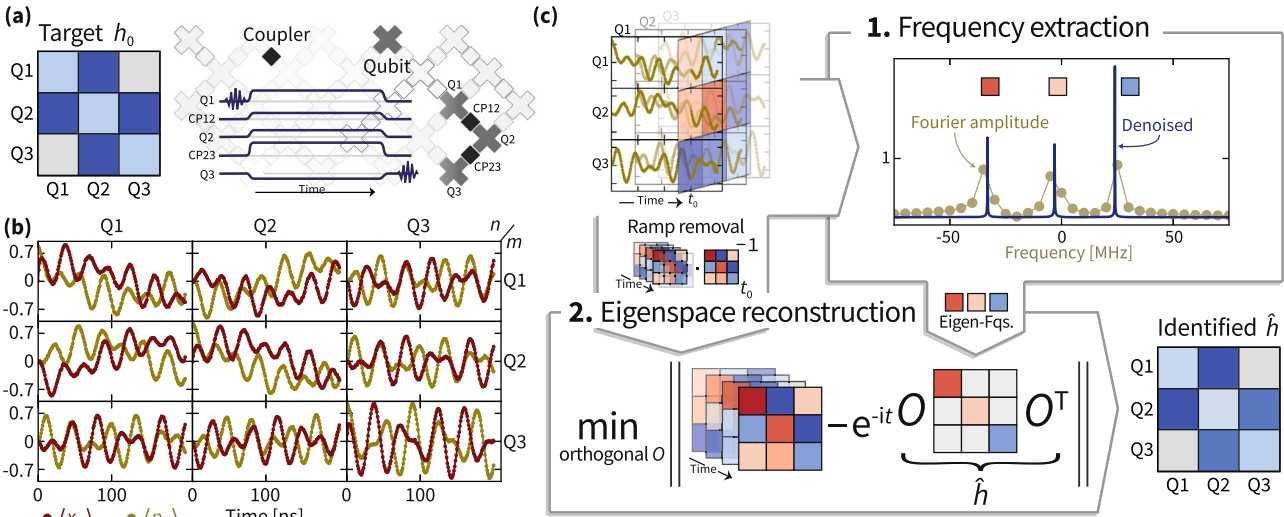

**Fig. 1 | Outline of the experiment and identification algorithm. a** The time evolution under a target Hamiltonian $h_0$ is implemented on an part of the Google Sycamore chip (gray) using the pulse sequence depicted in the middle. **b** The expected value of canonical coordinates $x_m$ and $p_m$ for each qubit $m$ over time is estimated from measurements using different $\psi_n$ as input states. **c** The data shown in (**b**) for each time $t_0$ can be interpreted as a (complex-valued) matrix with entries indexed by measured and initial excited qubit, $m$ and $n$. The identification algorithm proceeds in two steps: 1. From the matrix time-series, the Hamiltonian eigenfrequencies are extracted using our newly introduced algorithm coined

*tensorESPRIT*, introduced in the Supplemental Material, or an adapted version of the ESPRIT algorithm. The blue line indicates the denoised, high-resolution signal as "seen" by the algorithm. 2. After removing the initial ramp using the data at some fixed time, the Hamiltonian eigenspaces are reconstructed using a non-convex optimization algorithm over the orthogonal group. We obtain a diagonal orthogonal estimate of the final ramp. From the extracted frequencies and reconstructed eigenspaces, we can calculate the identified Hamiltonian $\hat{h}$ that describes the measured time evolution and a tomographic estimate of the initial ramp.

applied to one of the qubits, resulting in its Bloch vector moving to the equator. Then ramping pulses are applied to all qubits to bring them to the desired detuning around a common rendezvous frequency (6500 MHz in this work). At the same time, pulses are applied to the couplers to set the nearest-neighbor hopping to the desired value (20 MHz in this work). The pulses are held at the target values for time $t$, corresponding to the evolution time of the experiment. Subsequently, the couplers are ramped back to zero coupling and the qubits back to their initial frequency, where $\langle x_m(t)\rangle$ and $\langle p_m(t)\rangle$ on the desired qubit $m$ is measured. The initial and final pulse ramping take place over a finite time of 2–3 ns, and therefore give rise to a non-trivial effect on the dynamics, which we take into account in the identification procedure. In fact, we find that the effects of the ramping phase are the dominant source of SPAM errors in the quench-based analog simulation. The experimental data (Fig. 1b) on $N$ qubits are $N \times N$ time-series estimates of $\langle a_m(t)\rangle_{\psi_n}$ for $t = 0, 1, ..., T$ ns and all pairs $n, m = 1, ..., N$. Given those data, the identification task amounts to identifying the "best" coefficient matrix $h$, describing the time-sequence of snapshots of the single-particle unitary matrix $\frac{1}{2}\exp(-ith)$.

### Identification method
We can identify the generator $h$ of the unitary in two steps (Fig. 1c), making use of the eigendecomposition of the Hamiltonian (see "Methods"). In the first step, the time-dependent part of the identification problem is solved, namely, identifying the Hamiltonian eigenvalues (eigenfrequencies). In the second step, given the eigenvalues, the eigenbasis for the Hamiltonian of $h$ is determined. In order to make the identification method noise-robust, we furthermore exploit structural constraints of the model. First, the Hamiltonian has a spectrum such that the time-series data has a time-independent, sparse frequency spectrum with exactly $N$ contributions. Second, the Fourier coefficients of the data have an explicit form as the outer product of the orthogonal eigenvectors of the Hamiltonian. Third, the Hamiltonian parameter matrix is real and has an a priori known sparse

support due to the experimental connectivity constraints. These structural constraints are not respected by various sources of incoherent noise, including particle loss and finite shot noise, and coherent noise, in particular the SPAM error. Thereby, an identification protocol that takes these constraints into account is intrinsically robust against various imperfections. Importantly, we do not assume that the dynamics of the device is completely governed by a non-interacting, particle-number preserving Hamiltonian of the form (2). We rather impose this as a constraint on the reconstructed Hamiltonian and, as such, identify the best-fit Hamiltonian satisfying the constraint. Our approach thus robustly identifies the non-interacting part of a potentially interacting system.

To robustly identify the sparse frequencies from the experimental data, we develop a new super-resolution and denoising algorithm tensorESPRIT that is applicable to matrix-valued time series and uses tensor network techniques in conjunction with super-resolution techniques for scalar data[55]. Achieving high precision in this step is crucial for identifying the eigenvectors in the presence of noise. To robustly identify the eigenbasis, in the second step, we perform least-square optimization of the time-series data under the orthonormality constraint with a gradient descent algorithm on the manifold structure of the orthogonal group[57]. Additionally, we can incorporate the connectivity constraint on the coefficient matrix $h$ by making use of regularization techniques[61].

### Robustness against ramp errors
The initial and final ramping pulses result in a time-independent, linear transformation at the beginning and end of the time series. It is important to stress that such ramping pulses are expected to be generic in a wide range of experimental implementations of dynamical analog quantum simulations. Robustness of a Hamiltonian identification method against these imperfections is essential for accurate estimates in practice. We can model the effect of such *state preparation and measurement* (SPAM) errors via linear maps $S$ and $M$, which

alters our model of the ideal data (3) to

$$\langle a_m(t) \rangle_{\psi_n} = \frac{1}{2}(M \cdot \exp(-ith) \cdot S)_{m,n}. \tag{4}$$

These linear maps capture the effect of particle-number preserving quenches, as well as the projection of more general channels to the single-particle subspace. Any deviation of the observed experimental dynamics from our model of the data (4) will be visible in the quality of fit.

While for the frequency identification such time-independent errors "only" deteriorate the signal-to-noise ratio, for the identification of the eigenvectors of $h$ it is crucial to take the effects of non-trivial $S$ and $M$ into account. Given the details of the ramping procedure, we expect that the deviation of the initial map $S$ from the identity will be significantly larger than that of the final map $M$ and provide evidence for this in the Methods. In particular, the final map will be dominated by phase accumulation on the diagonal.

By pre-processing the data, we can robustly remove an arbitrary initial map $S$. By post-processing, we can obtain an orthogonal diagonal estimate $\hat{D}_M$ of the final map $M$. We give numerical evidence that the estimate $\hat{D}_M$ gives good results in the particular experimental setting. From the identified Hamiltonian and an orthogonal diagonal estimate $\hat{D}_M$ of $M$, we get an estimate $\hat{S}$ of $S$.

### Error sources

There are two main remaining sources of error that affect the Hamiltonian identification. First, the estimate $\hat{h}$ has a statistical error due to the finite number of measurements used to estimate the expectation values. Second, any non-trivial final map $M$ will produce a systematic error in the eigenbasis reconstruction and the tomographic estimate $\hat{S}$. We partially remedy this effect with an orthogonal diagonal estimate $\hat{D}_M$ of $M$.

### Predictive power

If the dynamics of the device is indeed coherent and particle-number preserving, the learned model will allow us to accurately predict the dynamics of the device in the single-particle subspace. If, additionally, interactions are negligible, the predictive power of our model extends to dynamics of more particles. This allows us to benchmark the Sycamore chip as a programmable quantum simulator of the non-interacting part of a Bose-Hubbard model. Accurately predicting the dynamics of many particles requires a generalization of our method to at least the two-particle sector.

### Experimental implementation

We implement and characterize different Hamiltonians from time-series data on two distinct quantum Sycamore processors—Sycamore #1 and #2. The Hamiltonians we implement have a fixed overall hopping strength $J_{i,j} = 20$ MHz and site-dependent local potentials $\mu_i$ on subsets of qubits. Specifically, we choose the local potentials quasi-randomly $\mu_q = 20\cos(2\pi qb)$ MHz, for $q = 1, \ldots, N$, where $b$ is a number between zero and one. In one dimension, this choice corresponds to implementing the Harper Hamiltonian, which exhibits characteristic "Hofstadter butterfly" frequency spectra as a function of the dimensionless magnetic flux $b$[62].

We measure deviations in the identification in terms of the *analog implementation error* of the identified Hamiltonian $\hat{h}$ with respect to the targeted Hamiltonian $h_0$ as

$$\mathcal{E}_{\text{analog}}(\hat{h}, h_0) := \frac{1}{N} \left\| \hat{h} - h_0 \right\|_{\ell_2}, \tag{5}$$

defined in terms of the $\ell_2$-norm, which for a matrix $A$ is given by $\|A\|_{\ell_2} = (\sum_{i,j} |A_{i,j}|^2)^{1/2}$. We also use the analog implementation error to

quantify the implementation error of the initial map $\hat{S}$ as $\mathcal{E}_{\text{analog}}(\hat{S}, \mathbb{1})$, and of the eigenfrequencies eig($\hat{h}$) as $\mathcal{E}_{\text{analog}}(\text{eig}(\hat{h}), \text{eig}(h_0))$. Notice that the analog implementation error of the frequencies in the data from the targeted Hamiltonian eigenfrequencies give a lower bound to the overall implementation error of the identified Hamiltonian. This is because the $\ell_2$-norm used in the definition (5) of $\mathcal{E}_{\text{analog}}$ is unitarily invariant and any deviation in the eigenbasis, which we identify in the second step of our algorithm, will tend to add up with the frequency deviation.

In Fig. 2, we illustrate the properties of a single Hamiltonian identification instance in terms of both how well the simulated time evolution fits the experimental data (a,d,e) and how it compares to the targeted Hamiltonian (b) and SPAM (c). We find that most entries of the identified Hamiltonian deviate from the target Hamiltonian by less than 0.5 MHz with a few entries deviating by around 1–2 MHz. The overall implementation error is around 1 MHz. The error of the identification method is dominated by the systematic error due to the final ramping phase that is around 1 MHz for the individual entries, see the Supplemental Material for details. Small long-range couplings exceeding the statistical error are necessary to fit the data well even when penalizing those entries via regularization. These entries are rooted in the effective rotation by the final ramping before the measurement and within the estimated systematic error.

The fit deviation from the data (Fig. 2e) exhibits a prominent decrease within the first few nanoseconds of the time evolution. This indicates that the time evolution differs during the initial phase of the experiment as compared to the main phase of the experiment, which we can attribute to the initial pulse ramping of the experiment. The identified initial map describing this ramping (Fig. 2c) is approximately band-diagonal and deviates from being unitary, indicating fluctuations of the effective ramps between different experiments.

We find a larger time-averaged real-time error (Fig. 2d) in all data series $\langle a_m \rangle_{\psi_n}$ in which $Q_4$ was measured, indicating a measurement error on $Q_4$. We also observe a deviation between the parameters of the target and identified Hamiltonian in qubits $Q_3$ and $Q_4$ and the coupler between them. Since the deviation of the eigenfrequencies is much smaller than of the full Hamiltonian, we attribute those errors also to a non-trivial final ramping phase at those qubits that leads to a rotated eigenbasis.

In Fig. 3, we summarize multiple identification data of this type to benchmark the overall performance of a fixed set of qubits. In panel (a), we show the measured Fourier domain data for 51 different values of the magnetic flux $b \in [0, 1]$. In panel (b), we plot the deviation of the frequencies identified from the data. Most implemented frequencies deviate by less than 1 MHz from their targets. Importantly, the frequency identification is robust against systematic measurement errors. When comparing the analog implementation errors of the full Hamiltonian (Fig. 3c) to the corresponding frequency errors, we find an up to fourfold increase in implementation error. The Hamiltonian implementation error is affected by a systematic error due to the non-trivial final ramp. We estimate this error using a linear ramping model; see the Supplemental Material for details. Since the deviation lies outside of the combined systematic and statistical error bars, our results indicate that the targeted Hamiltonian has not been implemented exactly.

In Fig. 3d, we show the median of the entry-wise deviation of the identified Hamiltonian from its target over all magnetic flux values. Thereby, the ensemble of Hamiltonians defines an overall error benchmark. This benchmark can be associated to the individual constituents of the quantum processor, namely, the qubits, corresponding to diagonal entries of the Hamiltonian deviation, and the couplers, corresponding to the first off-diagonal matrix entries of the deviation.

We use this benchmark over an ensemble of two flux values to assess a 27-qubit array of superconducting qubits. To do so, we repeat the analysis reported in Fig. 3 for 5-qubit dynamics on different subsets

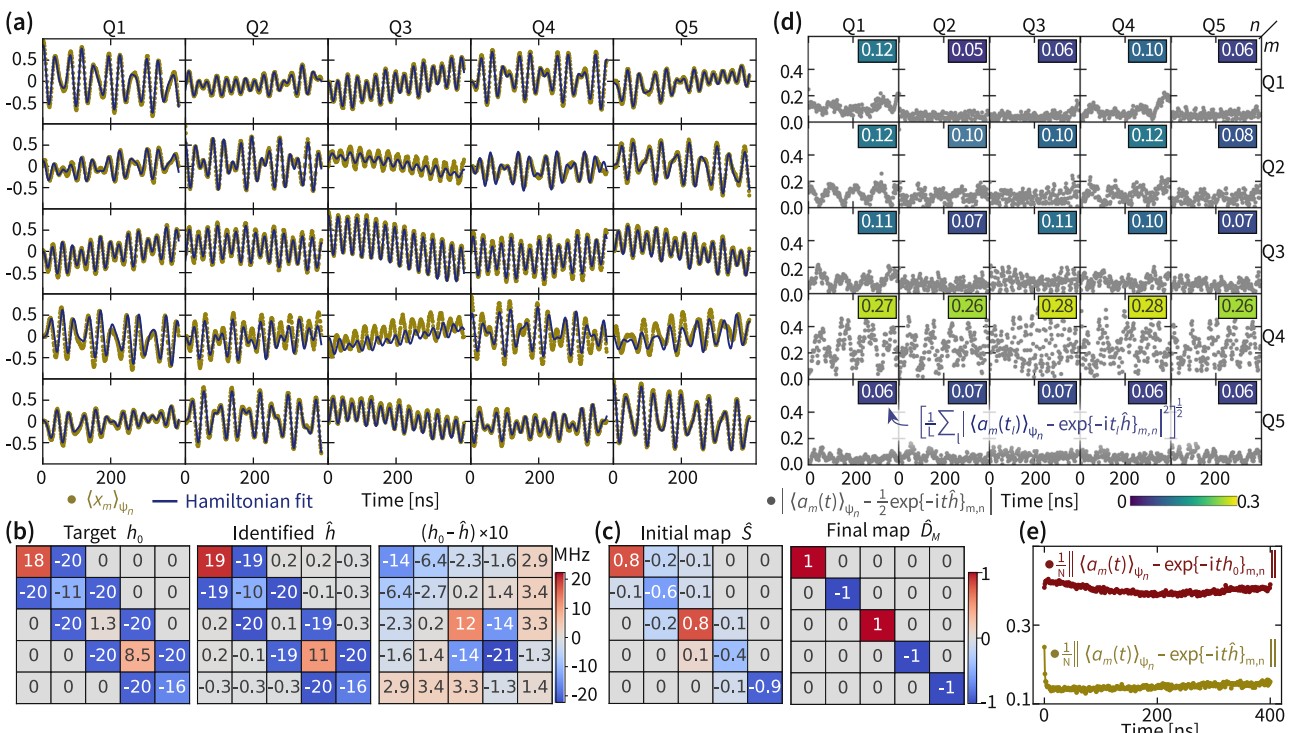

**Fig. 2 | A single Hamiltonian recovery of a 5-mode Hamiltonian and the corresponding time domain data. a** The full experimental time-series data $\langle x_m(t)\rangle_{\psi_n}$ for $m, n = 1, \ldots, 5$ and the best fit of those data in terms of our model $\frac{1}{2}(M \exp(-\mathrm{i}th)S)_{m,n}$ for a diagonal and orthogonal $M$ and linear map $S$ (solid lines). **b** The target Hamiltonian matrix $h_0$, the identified Hamiltonian $\hat{h}$, and the deviation between them. The error of each diagonal entry is $\pm(0.16 + 0.99)$ MHz and of each off-diagonal entry $\pm(0.12 + 0.50)$ MHz and comprises of the statistical and the systematic error, respectively. The analog implementation error $\mathcal{E}_{\text{analog}}(\hat{h}, h_0)$ is $0.73 \pm (0.07 + 0.62)$ MHz, and $0.32 \pm 0.00$ MHz for the eigenfrequencies. The analog implementation error $\mathcal{E}_{\text{analog}}(\hat{S}, \mathbb{1})$ of the identified initial map is $0.61 \pm (0.00 + 0.12)$. **c** The real part of the initial map $\hat{S}$ and the diagonal orthogonal estimate $\hat{D}_M$ of the final map $M$, inferred from the data using the identified Hamiltonian $\hat{h}$. **d** Absolute value of the time-domain deviation of the fit from the full experimental data for each time series, given by deviation$[\hat{h}, \hat{S}, \hat{D}_M](t)_{m,n} := \langle a_m(t)\rangle_{\psi_n} - \frac{1}{2}\hat{D}_M \exp(-\mathrm{i}t\hat{h})\hat{S}$. The insets represent the root-mean-square deviation of the Hamiltonian fit from the experimental data per time series, averaged over the evolution time for each matrix entry $(m, n)$, resulting in an entry-wise summarized quality of fit. We find a total root-mean-square deviation of the fit of 0.14. **e** Instantaneous root-mean-square deviation of the identified Hamiltonian $\hat{h}$, initial map $\hat{S}$ and final map $\hat{D}_M$ and of the target Hamiltonian $h_0$ with initial map fit $S_0$ from the experimental data averaged over the distinct time series.

of qubits and extract average errors of the individual qubits and couplers involved in the dynamics, both in terms of the identified Hamiltonian and the initial and final maps. Summarized in Fig. 4, we find significant variation in the implementation error of different couplers and qubits. While for some qubits the effects of the initial and final maps are negligible, for others they indicate the potential of a significant implementation error. From a practical point of view, such diagnostic data allows to maximally exploit the chip's error for small-scale analog simulation experiments. Let us note that within the error of our method the overall benchmark for the qubits and couplers for 5-qubit dynamics agrees with that of 3- and 4-qubit dynamics.

All of the Hamiltonian identification experiments discussed so far (Figs. 2, 4) were implemented on the Sycamore #1 chip. In order to compare these results to implementations on a physically distinct chip with different calibration, and to demonstrate the scalability of our method, we implement Hamiltonian identification experiments for an increasing number of qubits on the Sycamore #2 chip. More precisely, for a given number of qubits $N$, we implement many different Hamiltonians with quasi-random local potentials, as shown in Fig. 3c for $N = 6$. We then average the analog implementation errors of the Hamiltonians and frequencies for several system sizes. The results are shown in Fig. 5. Notably, comparing the two different processors, the overall quality of fit does not depend significantly on either the number of qubits or the processor used. This indicates, first, that our reconstruction method works equally well in all scenarios and, second, that both quantum processors implement Hamiltonian time evolution

that closely fits our model assumption. We also notice that the overall analog implementation error does not significantly depend on the system size. This signifies that no additional non-local errors are introduced into the system as the size is increased. At the same time, the overall error of Hamiltonian implementations on Sycamore #2 is much worse compared to those on Sycamore #1, indicating that Sycamore #2 was not as well calibrated. Hamiltonian identification thus allows us to meaningfully compare Hamiltonian implementations across different physical systems and system sizes.

## Discussion

We have implemented analog simulation of the time-evolution of non-interacting bosonic Hamiltonians with tunable parameters for up to 14 qubit lattice sites. A structure-exploiting learning method allows us to robustly identify the implemented Hamiltonian that governs the time-evolution. To achieve this, we have introduced a new super-resolution algorithm, referred to as tensorESPRIT, for precise robust identification of eigenfrequencies of a Hermitian matrix from noisy snapshots of the one parameter unitary subgroup it generates. Thereby, we diagnose the deviation from the target Hamiltonian and assess the accuracy of the implementation. We achieve sub-MHz error of the Hamiltonian parameters compared to their targeted values in most implementations. Combining the average performance measures over ensembles of Hamiltonians we associate benchmarks to the components of the superconducting qubit chips that quantify the performance of the hardware on the time evolution and provide specific

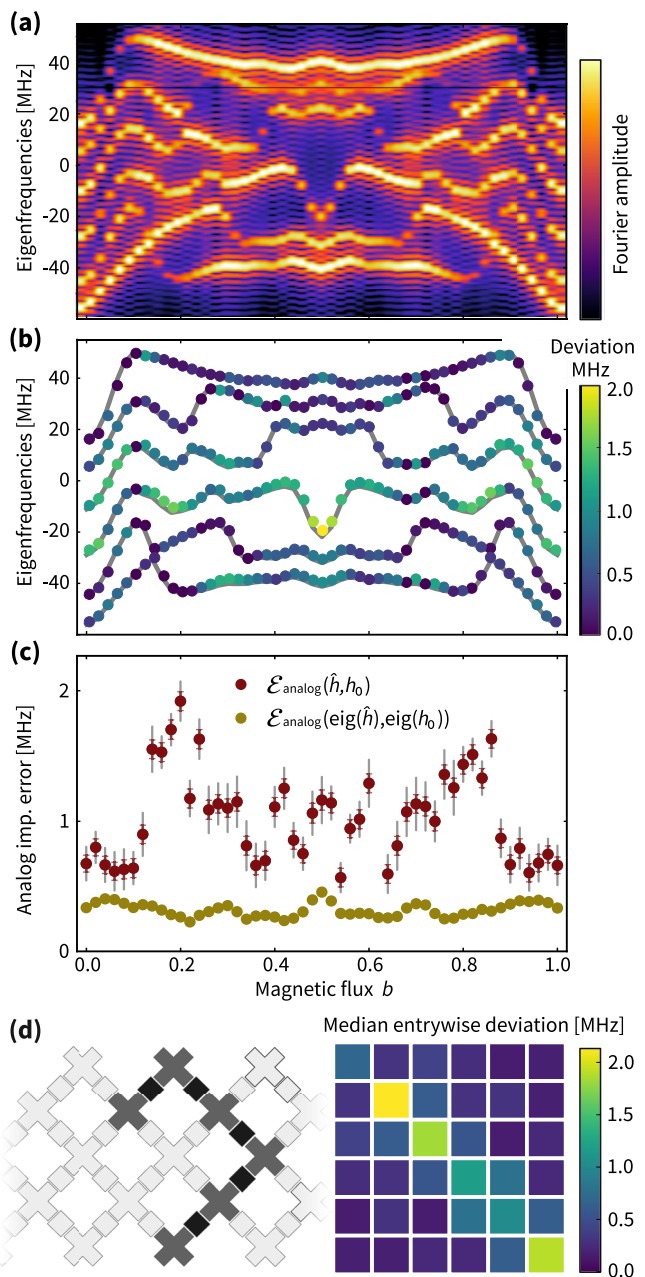

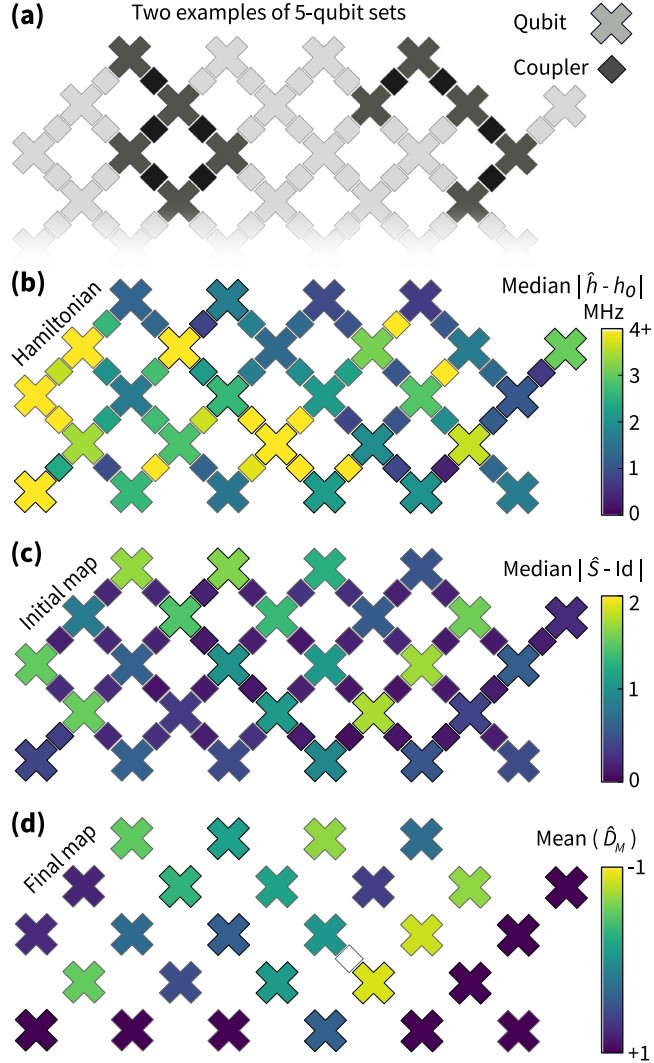

**Fig. 4 | Error map of Hamiltonian implementation across the Sycamore processor.** Over the grid of 27 qubits, we randomly choose subsets of connected qubits and couplers of size $N = 5$. On each subset we implement two Hamiltonians with $b = 0, 0.5$ and run the identification algorithm. Two instances are shown in (**a**). For each subset, we compute the deviation of the identified Hamiltonian and initial map from their respective target and assign it to each qubit or coupler involved. Due to overlap of subsets, each qubit or coupler has been involved in at least five different choices of subsets. **b**, **c** Show the median deviation for the Hamiltonian and initial map implementations, respectively. **d** Shows the mean of the sign flips in the identified (diagonal ± 1) final map for each qubit.

**Fig. 3 | Comparing frequency and full identification errors. a** In an $N = 6$ subset of connected qubits, by varying $b$ from 0 to 1, we implement 51 different Hamiltonians. The plot shows the Fourier transform of the time domain data. **b** The extracted eigenfrequencies (denoised peaks in (**a**)) are shown as colored dots, where the assigned color is indicative of the deviation between targeted eigenfrequencies (gray lines) and the identified ones from position of the peaks. **c** Analog implementation error $\mathcal{E}_{analog}(\hat{h}, h_0)$ of the identified Hamiltonian (dark red) compared to the implementation error $\mathcal{E}_{analog}(\mathrm{eig}(\hat{h}), \mathrm{eig}(h_0))$ of the identified frequencies (golden). Colored (gray) error bars quantify the statistical (systematic) error. **d** Layout of the six qubits on the Sycamore processor and median of the entry-wise absolute-value deviation of the Hamiltonian matrix entries from their targeted values across the ensemble of 51 different values of $b \in [0, 1]$.

diagnostic information. Within our Hamiltonian identification framework, we are able to identify SPAM errors due to parameter ramp phases as a severe limitation of the architecture. Importantly, such ramp phases are present in any analog quantum simulation of quenched dynamics. Our results show that minimizing those is crucial for accurately implementing a Hamiltonian.

The experimental and computational effort of the identification method scales efficiently in the number of modes of the Hamiltonian. We have also numerically identified the limitations of more direct algorithmic approaches and demonstrated the scalability of our method under empirically derived noise and error models.

We have demonstrated and custom-tailored our approach here to a superconducting analog quantum simulation platform. It can be applied directly to any bosonic and fermionic analog simulation platform which allows for accurate preparation and measurement of single particle excitations at specific lattice sites. Generalizing our two-step approach developed here, we expect a polynomial scaling with the dimension of the diagnosed particle sector and therefore remain efficient for diagnosing two-, three- and four-body interactions, thus allowing to build trust in the correct implementation of interacting Hamiltonian dynamics as a whole. Furthermore, it is in some cases

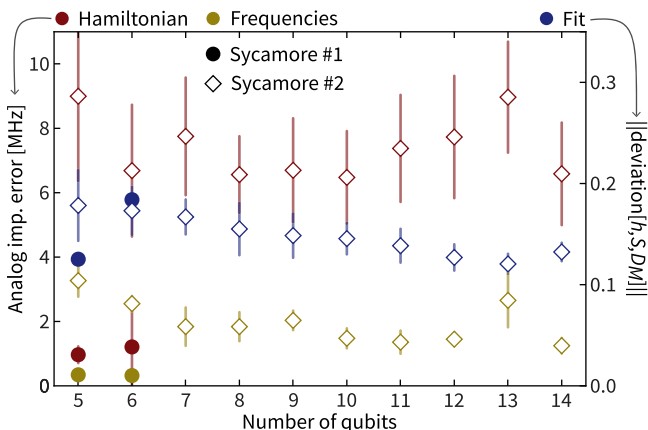

**Fig. 5 | Analog implementation error scaling and comparing different quantum processors.** We measure the analog implementation error of the implemented Hamiltonians (dark red) and their eigenfrequencies (golden) as well as the deviation $(\sum_{l=0}^{L} \| \text{deviation}[\hat{h}, \hat{S}, \hat{D}_M](t_l)\|_{\ell_2}^2 /(N^2(L+1)))^{1/2}$ of the fit from the experimental data (dark blue) all averaged over implementations of Hamiltonians with quasi-random local potential on an increasing number of qubits on two different quantum processors—Sycamore #1 (circles) and #2 (diamonds). Each point is the mean of the respective quantity over 51 Hamiltonian implementations (21 for $N = 5$ and 20 for $N = 14$ on Sycamore #2). The data points at $N = 6$ on Sycamore #1 summarizes Fig. 3c. The error bars represent one standard deviation.

possible to adapt the method to Hamiltonians with general non-particle number preserving free part. From a broader perspective, with this work, we hope to contribute to the development of a machinery for precisely characterizing and thereby improving analog quantum devices.

## Methods

### Experimental details

**Details on the quantum processor.** We use the Sycamore quantum processor composed of quantum systems arranged in a two-dimensional array. This processor consists of gmon qubits (transmons with tunable coupling) with frequencies ranging from 5 to 7 GHz. These frequencies are chosen to mitigate a variety of error mechanisms such as two-level defects. Our coupler design allows us to quickly tune the qubit-qubit coupling from 0 to 40+ MHz. The chip is connected to a superconducting circuit board and cooled down to below 20 mK in a dilution refrigerator. The median values of the $T_1$ and $T_2$ times of the qubits are $T_1 = 16.1 \, \mu s$, $T_2 = 5.3 \, \mu s$ (Ramsey interferometry) and $T_2 = 17.8 \, \mu s$ (after CPMG dynamical decoupling). Each qubit has a microwave control line used to drive an excitation and a flux control line to tune the frequency. The processor is connected through filters to room-temperature electronics that synthesize the control signals. We execute single-qubit gates by driving 25 ns microwave pulses resonant with the qubit transition frequency, resulting in single-qubit gate fieldity of 99.8% as measured via randomized benchmarking.

**Ramping pulses.** The pulses used in the experiment are pre-distorted in order to compensate for filters on the control lines. In order to calibrate this distortion, we send rectangular pulses to the qubits and monitor the frequency change of the qubits. This allows us to know the response of the microwave lines at the qubits (i.e., the deviation from a rectangle) and compensate for distortions. The ramp time can be as fast as 2–3 ns and the distortions take the form of overshoot and undershoots with a long response time of 100ns. After compensating for the distortions, the qubit frequency remains fixed.

**Experimental read-out and control.** The qubits are connected to a resonator that is used to read out the state of the qubit. The state of all

qubits can be read simultaneously by using a frequency-multiplexing. Initial device calibration is performed using "Optimus"[63] where calibration experiments are represented as nodes in a graph.

### Details of the identification algorithm

Succinctly written, our data model is given by

$$y_{m,n}[l] := \langle a_m(t_l)\rangle_{\psi_n} = \frac{1}{2}(M \cdot \exp(-it_l h) \cdot S)_{m,n}, \quad (6)$$

where $m, n = 1, ..., N$ label the distinct time series, $l = 0, ..., L$ labels the time stamps of the $L+1$ data points per time series. The matrices $S$ and $M$ are arbitrary invertible linear maps that capture the state preparation and measurement stage, as affected by the ramping of the eigenfrequencies of the qubits and couplers to their target value and back (see Fig. 1). In the experiment, we empirically estimate each such expectation value with 1000 single shots.

Our mindset for solving the identification problem is based on the eigendecomposition $h = \sum_{k=1}^{N} \lambda_k |v_k\rangle\langle v_k|$ of the coefficient matrix $h$ in terms of eigenvectors $|v_k\rangle$ and eigenvalues $\lambda_k$. We can write the data (6) in matrix form as

$$y[l] = \frac{1}{2}\exp(-it_l h) = \frac{1}{2}\sum_{k=1}^{N} e^{-it_l\lambda_k}|v_k\rangle\langle v_k|, \quad (7)$$

where we have dropped $S$ and $M$ for the time being. This decomposition suggests a simple procedure to identify the Hamiltonian using Fourier data analysis. From the matrix-valued time series data $y[l]$ (7), we identify the Hamiltonian coefficient matrix $h$ in two steps. First, we determine the eigenfrequencies of $h$. Second, we identify the eigenbasis of $h$. To achieve those identification tasks with the largest possible robustness to error, it is key to exploit all available structure at hand.

**Step 1: Frequency extraction.** In order to robustly estimate the spectrum, we exploit that the signal is sparse in Fourier space. This structure allows us to substantially denoise the signal and achieve super-resolution beyond the Nyquist limit[64,65]. A candidate algorithm for this task, suitable for scalar time-series, is the ESPRIT algorithm, which comes with rigorous recovery guarantees[55,56]. To extract the Hamiltonian spectrum from the matrix time-series $y[l]$, we apply ESPRIT to the trace of the data series (for $S = M = \mathbb{1}$)

$$F[l] := \text{Tr}[y[l]] = \sum_{m=1}^{N} y_{m,m}[l] = \frac{1}{2}\sum_{k=1}^{N} e^{-it_l\lambda_k}. \quad (8)$$

The drawback of this approach is that if the spectrum of the Hamiltonian is sufficiently crowded, which will happen for large $N$, the Fourier modes in $F[l]$ become indistinguishable and ESPRIT fails to identify the frequencies. In particular, ESPRIT is not able to identify degeneracies in the spectrum.

To overcome this issue and obtain a truly scalable learning procedure applicable to degenerate spectra, we develop a new algorithm coined tensorESPRIT. TensorESPRIT extends the ideas of ESPRIT to the case of matrix-valued time series using tensor network techniques. Using tensorESPRIT also improves the robustness of frequency estimation to SPAM errors. For practical Hamiltonians, tensorESPRIT becomes necessary for systems with $N \gtrsim 12$; as we find in numerical simulations summarized in Section C and detail in Section IV.B of the Supplemental Material.

TensorESPRIT (ESPRIT) comprises of a denoising step, in which the rank of the Hankel tensor (matrix) of the data is limited to its theoretical value. Subsequently, rotational invariance of the data is used to compute a matrix from the denoised Hankel tensor (matrix), the spectrum of which has a simple relation to the spectrum of $h$. In the case of ESPRIT, this amounts to a multiplication of the denoised Hankel

matrix by a pseudoinverse of its shifted version. Contrastingly, tensorESPRIT uses a sampling procedure to contract certain sub-matrices of the denoised Hankel tensor with the pseudoinverse of other sub-matrices. Details on both algorithms can be found in the Supplemental Material.

**Step 2: Eigenspace identification.** To identify the eigenspaces of the Hamiltonian, we use the eigenfrequencies found in Step 1 to fix the oscillating part of the dynamics in Eq. (7). What remains is the problem of finding the eigenspaces $|v_k\rangle\langle v_k|$ from the data. This problem is a non-convex inverse quadratic problem, subject to orthogonality of the eigenspaces, as well as the constraint that the resulting Hamiltonian matrix respects the connectivity of the superconducting architecture. Formally, we denote the a priori known support set of the Hamiltonian matrix as $\Omega$, so that we can write the support constraint as $h_{\bar{\Omega}} = 0$, where $\bar{\Omega}$ denotes the complement of $\Omega$ and subscripting a matrix with a support set restricts the matrix to this set. We can cast this problem into the form of a least-squares optimization problem

$$\underset{\{|v_k\rangle\}}{\text{minimise}} \qquad \sum_{l=0}^{L} \left\| y[l] - \sum_k e^{-i\lambda_k t_l}|v_k\rangle\langle v_k| \right\|_{\ell_2}^2,$$

$$\text{subject to} \quad \langle v_m|v_n\rangle = \delta_{m,n} \quad \text{and} \quad \left(\sum_k \lambda_k|v_k\rangle\langle v_k|\right)_{\bar{\Omega}} = 0,$$
(9)

equipped with non-convex constraints enforcing orthogonality, and the quadratic constraint restricting the support. In order to approximately enforce the support constraint, we make use of regularization[61]. It turns out that this can be best achieved by adding a term [[66], App. A]

$$\mu \left\| \left(\sum_k \lambda_k|v_k\rangle\langle v_k|\right)_{\bar{\Omega}} \right\|_{\ell_2}$$
(10)

to the objective function (9), where $\mu > 0$ is a parameter weighting the violation of the support constraint. We then solve the resulting minimization problem by using a conjugate gradient descent on the manifold of the orthogonal group[57,67], see also the recent work[68–70] for the use of geometric optimization for quantum characterization.

Without the support constraint this gives rise to an optimization algorithm that converges well, as shown in the Supplemental Material. However, the regularization term makes the optimization landscape rugged as it introduces an entry-wise constraint that is skew to the orthogonal manifold. To deal with this, we consecutively ramp up $\mu$ until the algorithm does not converge anymore in order to find the Hamiltonian that best approximates the support constraint while being a proper solution of the optimization problem. For example, for the data in Fig. 2 the value of $\mu$ is 121. In order to avoid that we identify a Hamiltonian from a local minimum of the rugged landscape, we only accept Hamiltonians that achieve a total fit of the experimental data within a 5% margin of the fit quality of the unregularized recovery problem, and use the Hamiltonian recovered without the regularization otherwise.

**Robustness to state preparation and measurement errors**
The experimental design requires a ramping phase of the qubit and coupler frequencies from their idle location to the desired target Hamiltonian and back for the measurement. In effect, the data model (6) includes time-independent linear maps $M$ and $S$ that are applied at the beginning and end of the Hamiltonian time-evolution. The maps affect both the frequency extraction and the eigenspace reconstruction.

For the frequency extraction using ESPRIT, the Fourier coefficients of the trace signal $F[l]$ become $\langle v_k|SM|v_k\rangle$. While the frequencies remain unchanged the Fourier coefficients now deviate from unity, significantly impairing the noise-robustness of the frequency

identification. This effect is still present, albeit weaker, in tensorESPRIT, in the case of non-unitary SPAM errors. The eigenspace reconstruction is affected much more severely and requires careful consideration, as detailed below and in the Supplemental Material.

**Ramp removal via pre-processing.** We can remove either the initial map $S$ or the final map $M$ from the data. To remove $S$, we apply the pseudoinverse $(\cdot)^+$ of the data $y[l_0]$ at a fixed time $t_{l_0}$ to the entire (time-dependent) data series in matrix form. For invertible $S$ and $M$ this gives rise to

$$y^{(l_0)}[l] = y[l](y[l_0])^+ = \sum_{k=1}^{N} e^{-i\lambda_k(t_l-t_{l_0})}M|v_k\rangle\langle v_k|M^{-1}.$$
(11)

The caveat of this approach is that the shot noise that affected the single time point $y[l_0]$ can lead to correlated errors in every entry of the new data series $y^{(l_0)}$.

We can reduce the error induced by these correlations by effectively averaging over "corrected" data series $y^{(l_0)}$ with different $l_0$. To this end, we compute the concatenation of data series for different choices of $l_0$, e.g., for every $s$ data points $0, s, 2s, ..., \lfloor L/s \rfloor s$ giving rise to new data $y_{\text{total},s} = (y^{(0)}, y^{(s)}, y^{(2s)}, \ldots, y^{(\lfloor L/s \rfloor s)}) \in \mathbb{C}^{\lfloor L/s \rfloor L}$. If the data suffers from drift errors, it is also beneficial to restrict each data series $y^{(l_0)}$ to entries $y^{(l_0)}[\kappa]$ with $\kappa \in [l_0 - w, l_0 + w]$, i.e., the entries in a window of size $w$ around $l_0$. In practice, we use $s = 1$ and $w = 50$ for the reconstructions on Sycamore #1, and $s = 1$, $w = L$ for those on Sycamore #2.

As we argue below, the final map $M$ is nearly diagonal here. Hence, we can use $y_{\text{total},s}$ from Eq. (9) and it is justified to apply the support constraint in the eigenspace reconstruction step. However, the eigenspace reconstruction will suffer from systematic errors due to the final map, even in the case when it is nearly diagonal. Below, we explain a method to partially remove this error.

**Tomographic estimate of $S$ and $M$.** The systematic error in the reconstructed Hamiltonian eigenbasis can be expressed as an orthogonal rotation $D_M$ from the eigenbasis that is actually implemented. Due to the gauge freedom in the model (6), we cannot hope to identify $D_M$ fully without additional assumptions. However, as elaborated on in the Supplemental Material, we can find a diagonal orthogonal estimate $\hat{D}_M$ of the true correction $D_M$ and hence remove a sign of the systematic error. To this end, we assume that the experimental implementation of the target Hamiltonian does not flip the sign in the hopping terms and remedy the sign of systematic error due to the final map by fitting a diagonal orthogonal rotation of the Hamiltonian eigenbasis $\hat{D}_M$ that minimizes the implementation error. We update the reconstructed Hamiltonian to

$$\hat{h} = \hat{D}_M \tilde{h} \hat{D}_M,$$
(12)

where $\tilde{h} = \sum_k \lambda_k|v_k\rangle\langle v_k|$ and $\{|v_k\rangle\}$ is the eigenbasis obtained by solving the problem (9), and use $\hat{D}_M$ as an estimate of $M$. We can now obtain a tomographic estimate of the initial map through

$$\hat{S} := \frac{2}{L+1}\sum_{l=0}^{L} \exp[it_l\hat{h}]\hat{D}_M y[l].$$
(13)

**Imbalance between initial and final ramping phase.** As explained above, the pre-processing step allow us to remove either the initial map $S$ or final map $M$ from the data, while we can only find a diagonal orthogonal estimate of the remaining map. A priori it is unclear which one of the two maps should be removed in order to reduce the systematic error more.

We have already treated the initial and final ramping phases on a different footing, however. The reason for this is rooted in the specifics of the ramping of the couplers compared to the qubits. The couplers

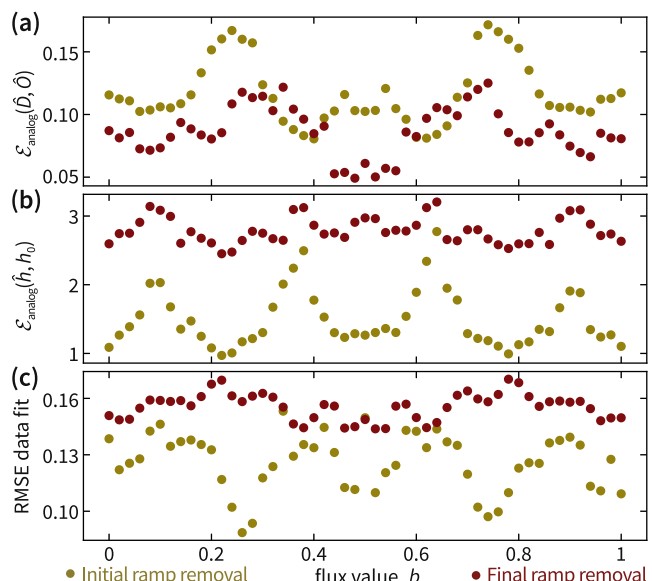

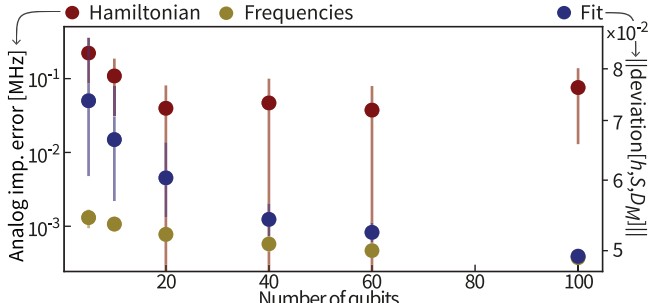

**Fig. 7 | Numerical benchmarks for larger system sizes.** Recovery error of frequencies (golden) and Hamiltonians (red) from simulated time series averaged over 20 instances of Harper Hamiltonians for different system sizes. The error bars represent one standard deviation. The evolution is simulated for up to 0.6 μs and sampled at a rate of 250 MHz. Statistical noise is simulated using $10^3$ shots per expected value and SPAM is modeled by using randomly chosen idle qubit and coupler frequencies, linear ramping of 1.5 GHz/s padded by 0.05 ns. The fitting error of the time series is depicted in blue, right y-axis. We refer to the Supplemental Material, Sec. VII A for details.

**Fig. 6 | Initial ramp removal versus final ramp removal.** We identify Hamiltonians of a set of 5-qubit Hamiltonians with Hofstadter butterfly potentials $\mu_q = 20\cos(2\pi qb)$ MHz for qubits $q = 1, \ldots, 5$ and flux value $b$ in without regularization. **a** Deviation of the orthogonal part $\hat{O}_S$ ($\hat{O}_M$) of the identified initial map $\hat{S}$ (final map $\hat{M}$) from the closest diagonal orthogonal matrix $\hat{D}_S$ ($\hat{D}_M$). **b** Analog implementation error of the corresponding identified Hamiltonians $\hat{h}_S$ ($\hat{h}_M$). **c** Total root-mean-square deviation of the time series data from the Hamiltonian fit.

need to be ramped from their idle frequencies to provide the desired target frequencies of 20 MHz. This is why we expect the time scale of the initial ramping to be mainly determined by the couplers, namely the delay until they arrive around the target frequency and the time it takes to stabilize at the target frequency. In contrast, the final ramping map becomes effectively diagonal as soon as the couplers are again out of the MHz regime. We therefore expect that the initial map has a sizeable non-diagonal orthogonal component, whereas the final map is approximately diagonal.

We build trust in this assumption using experimental data in Fig. 6. We observe that the deviation of the orthogonal part $\hat{O}_S$ of the identified initial map $\hat{S}$ from its projection $\hat{D}_S$ to diagonal orthogonal matrices is much larger than the corresponding deviation for the final map (Fig. 6a). Moreover, both the root-mean-square fit of the data (Fig. 6c) and the analog implementation error of the identified Hamiltonian with its target (Fig. 6b) are significantly improved when removing the initial ramp, as compared to removing the final ramp. This indicates that $S$ induces a larger systematic error than $M$. Correspondingly, it is indeed more advantageous to remove the initial map in the pre-processing and fit the final map with a diagonal orthogonal matrix, validating the approach taken here.

Overall, the recovered model ($\hat{h}, \hat{S}, \hat{D}_M$) fits the experimental data well, as demonstrated in Figs. 2, 5, 6, and gives good prediction accuracy on simulated data, as demonstrated in Fig. 7 in the next section as well as the Supplemental Material. In the Supplemental Material, we provide further numerical evidence that this approach leads to small systematic errors and recovers a model with good predictive power.

### Benchmarking the algorithm

We benchmark our identification algorithm against more direct approaches in numerical simulations including models for statistical and systematic errors in the Supplemental Material VI. We find that, indeed, already for small system sizes, the regularized manifold optimization algorithm developed here features an improved robustness against

state preparation and measurement errors compared to (post-projected) linear inversion. For intermediate system sizes ($N > 10$), exploiting structure in the recovery algorithm then becomes an imperative. In particular, for larger system sizes the eigenspectrum of the Hamiltonian becomes unavoidably narrower spaced, leading to (near-)degeneracies. We find that on instances of the Harper Hamiltonian studied here linear inversion approaches cannot be applied at all for $N > 20$. Regularized conjugate gradient decent in contrast yields good recovery performance even for larger systems. The same limitations apply to a direct Fourier analysis of the cumulative time series data using ESPRIT, as described above. For different families of Hamiltonians, we find that above a system size of $N \approx 20$ tensorESPRIT still consistently recovers the frequency spectrum, while the ESPRIT algorithm fails to do so.

Using structure not only allows our algorithm to denoise the data and achieve error robustness, it also makes precise Hamiltonian identification possible even with the number of measurements dramatically reduced in the spirit of compressed sensing. As described above, the number of measurements scales quadratically with the system size. We find that using the conjugate gradient algorithm the identification procedure reliably recovers Hamiltonians even when it has access to only about 3% of the measurements. In this regime, the linear inverse problem of finding the eigenvectors is underdetermined. Thus, the required experimental resources can be significantly reduced for large system sizes.

To demonstrate our method's scalability, Fig. 7 shows the recovery performance of the structure-exploiting algorithm on simulated data under realistic models for SPAM errors and with finite measurement statistics in the regime where the baseline approaches could not be applied anymore.

As detailed in the Supplemental Material, tensorESPRIT has computational complexity in $\mathcal{O}(L^2 N^3)$. It is not straight-forward to bound the computational complexity of the conjugate gradient descent, as it depends on the required precision of the matrix exponential and the number of descent steps until convergence. The entire identification algorithm consumes $\mathcal{O}(LN^2)$ memory. In practice, we find that the algorithm can be easily deployed on a consumer-grade laptop computer, e.g., reconstructing Hamiltonians of size $N = 50$ in around 5 min.

### Error estimation

We here discuss how we estimate the systematic and statistical contributions to the error on the identified Hamiltonian $\hat{h}$ and initial map $\hat{S}$. Note that the impact of the systematic error on predicting results of experiments with the same initial and final ramps is reduced due to the

gauge invariance of the model (6). Due to this freedom, some of the error in identifying $\hat{D}_M \sim M$ gets accounted for by a corresponding error in the identification $\hat{h} \sim h$ and $\hat{S} \sim S$ in expressions of the type $\hat{M}e^{-ith}\hat{S}$. This prediction error can be further decreased by running the algorithm twice—removing the initial map in the first run and the final map in the second run, using the first ramp estimates to partially remove the ramps from the data before running the second iteration of the identification. This procedure is detailed and supported by numerical evidence in the Supplemental Material.

**Systematic error: final ramp effect estimation.** In order to estimate the magnitude of the systematic error that is induced by the non-trivial final map, we use a linear model of the final ramping phase with a constant ramping speed and constant wait time between the coupler and qubit ramping. We detail and present validation of this ramping model with a separate experiment in the Supplemental Material, where we also provide empirical estimates of the model parameters.

Given a Hamiltonian matrix $\hat{h}$ and the initial ramp $\hat{S}$ obtained from experimental data, we recover the Hamiltonian matrix $\hat{h}'$ from data simulated using the model $(\hat{h}, \hat{S}, M)$, where $M$ is the final ramp given by our ramping model. We use $|f(\hat{h}) - f(\hat{h}')|$ as an estimate of the systematic error on quantities of the form $f(\hat{h}) \in \mathbb{R}$,

**Statistical error: bootstrapping.** We estimate the effect of finite measurement statistics on the Hamiltonian estimate that is returned by the identification method via parametric bootstrapping. To this end, we simulate time series data with statistical noise using Haar-random unitaries $S$ as initial ramps, the identified Hamiltonian $\hat{h}$ and final ramp $M = \mathbb{1}$, as detailed in the Supplemental Material.

## Data availability
The experimental data is available from the authors upon request.

## Code availability
The code is available from the authors upon request.

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

## Acknowledgements

We acknowledge contributions from Charles Neill, Kostyantyn Kechedzhi, and Alexander Korotkov to the calibration procedure used in this analog approach. We would like to thank Christian Krumnow, Benjamin Chiaro, Alireza Seif, Markus Heinrich, and Juani Bermejo-Vega for fruitful discussions in early stages of the project. The hardware used for this experiment was developed by the Google Quantum AI hardware team, under the direction of Anthony Megrant, Julian Kelly, and Yu Chen. D.H. acknowledges funding from the U.S. Department of Defense through a QuICS Hartree fellowship. This work has been supported by the BMBF (DAQC, MUNIQC-Atoms), for which it provides benchmarking tools for analog-digital superconducting quantum devices, as well as by the DFG (specifically EI 519 20-1 on notions of Hamiltonian learning, but also CRC 183 and GRK 2433 Deadalus). We have also received funding from the European Union's Horizon2020 research and innovation program (PASQuanS2) on programmable quantum simulators, the Munich Quantum Valley (K-8), the Einstein Foundation, Berlin Quantum, and the ERC (DebuQC).

## Author contributions

D.H. and I.R. conceived of the Hamiltonian identification algorithm. J.F. conceived of the tensorESPRIT algorithm. D.H., I.R., and J.F. analyzed the experimental data and benchmarked the identification algorithm. P.R. took the experimental data. D.H. and I.R. wrote the initial manuscript. D.H., I.R., J.F., J.E., and P.R. contributed to discussions and writing the final manuscript.

## Funding

## Competing interests

The authors declare no competing interests.
