## [Peer Review File · Nature Communications]

Robustly learning the Hamiltonian dynamics of a superconducting quantum processorREVIEWER COMMENTS

Reviewer #1 (Remarks to the Author):

In the submitted manuscript, the authors presented a way to robustly learn the free Hamiltonian parameters of a superconducting qubit analogue quantum simulator by measuring the single-mode canonical coordinates. The authors developed new estimation methods that are robust against state preparation and measurement (SPAM) errors. The authors demonstrated this algorithm on two Sycamore processors and achieved high estimation accuracy. Experimental Hamiltonian learning is a topic of increasing interest and involves efforts from both experimental and theoretical perspectives. The methods presented in the manuscript look promising and could stimulate follow-up research in Hamiltonian learning. Therefore, I would recommend the publication of this manuscript in Nature Communications after revisions to the methods.

Before its acceptance, since the main contribution of this work is an experimental proposal for free Hamiltonian learning, I would suggest the authors address my following concerns regarding the methods to strength the presentation and significance of this work.

In this work, the eigenstate of H lies in the single-excitation subspace, which is a strong constraint for the free Hamiltonian. As a result, the dimension of the system is just N instead of $\text{Poly}(N)$ in the case of more general free Hamiltonians. This consideration greatly decreases the difficulty of learning: in the frequency extractions, we only need to deal with N different frequencies, as indicated by Eq 7. Because of these strict constraints, I doubt whether the methods could be useful in relatively complicated cases. I'll elaborate on my concerns below.

1. When considering free Hamiltonians with many particles, more eigenfrequencies will emerge, and more importantly, the eigenspaces $|\nu_k\rangle\langle\nu_k|$ will become more complicated. I am curious if this method can deal with this case.
2. In continuation to the above point, the authors proposed to cast the eigenspace identification problem into a non-convex optimisation problem in Eq 9. Since $|\nu_k\rangle$ contains exponentially many elements, from my naïve understanding, this optimisation over $|\nu_k\rangle$ in Eq 9 is exponentially costly (though it might be simple for the case studied here). The authors may have mentioned the efficiency of the method, but this does not seem clear to me (especially when the target problem is not that simple, see point 3 as well). Could the authors comment on why the identification is efficient and what is the dependence on the dimension of the Hamiltonian?
3. I believe the above concern primarily stems from my limited understanding of the method in the context of a more general free-Hamiltonian case. I was wondering if learning the parameters requires some prior knowledge about the Hamiltonian parameters. For example, if the form of the Hamiltonian is unknown, could we still learn it? I suppose the answer is no. Even though the Hamiltonian is sparse (since the bosonic Hamiltonian is unbounded and may not be local in the qubit representations), this identification may still be hard.

There are not too many discussions on it. In addition, after getting the frequency λ_k , how to learn the parameters (given that v_k is unknown and/or the form of the Hamiltonian is unknown)? To answer my question, the authors do not need to present a method, but I just wanted to confirm what kind of problems can be done using the proposed method.

4. The essence of the frequency extraction, or more generally, the spectroscopic methods, is that we are able to observe the dynamics by measurement. In this context, the authors choose to create a single-quasiparticle excitation and measure on the x - p basis (local Pauli basis). However, for a relatively general Hamiltonian (even when it is noninteracting), it is in principle hard to guarantee that the signal associated with the initial state and the observable is non-zero (we could call it 'spectral weight', which is the initial state and the observable dependent). Could the authors comment on how to prepare the initial state and how to choose the observable (for either a simple or a general case)? Again, just some comments are expected.
5. The authors claimed that this TensorESPRIT can learn degenerate spectra. However, I did not find too much evidence in either Method or SI. I am a little confused about what the improvement over the existing approaches is.
6. In the discussion, the authors claimed, 'Generalising our two-step approach developed here, we expect a polynomial scaling with the dimension of the diagnosed particle sector and therefore remain efficient for diagnosing two-, three- and four-body interactions'. The focus of this paper is the identification of the non-interacting part. The generalisation is not that straightforward as the authors claimed.

A major result of this work is the robustness against SPAM errors, but I did not fully understand this point. It looks unclear how to avoid 'the noise that affected $y[0]$ is now present in every entry of the new data series'. Following Eq 11, the authors said that 'We can remedy this effect by using several time points $y[0]$ for the inversion. To this end, we compute the concatenation of data series for different choices of I_0 '. The authors may answer this in SI, but there is no explanation about the intuitions behind it at least in Methods. Could the authors comment on whether the case is much easier with SPOM error only.

I felt confused about why D_M is nearly orthogonal. When I looked at SI, I found that the authors referred to Method C: 'we expect the final ramp M to be nearly diagonal', but in Method C it said this will be discussed in detail in SI. I suppose this is a very important finding in this work, but it is not that obvious to me.

Could the authors explain why the deviation of the target Hamiltonian (red line) is much larger than that of the identified Hamiltonian (yellow line) in Figure 2. From my understanding, there should be no deviation of the target Hamiltonian in the noiseless case, but I may get wrong on this point.

I felt confused about the simulations in Figure 4. The authors said that ‘we repeat the analysis reported in Fig. 3 for 5-qubit dynamics on different subsets of qubits’. Does this mean the authors only estimated the Hamiltonians on at most 5 qubits (instead of 27 qubits)? If so, does it make sense to show the deviation of the Hamiltonian and initial maps for 27 qubits in Figure 3.

In Methods D the authors tested the algorithm for larger systems. I was wondering why the error becomes smaller with increasing system size in Figure 7. Does this align with our expectations?

I am curious about how the authors run the noisy simulation up to 100 qubits. It looks like this process with random unitaries and SPAM errors is not classically stimulable.

I am not an expert in commenting on the experimental details, but I am curious about why the energy scale is quite different from the author Roushan’s previous work [Science. 2017 Dec 1;358(6367):1175-1179. doi: 10.1126/science.aao1401.], which I believe is quite relevant to this manuscript. In figure 2 of the Science paper, the energy scale is about one order large than that in the submitted manuscript, while the spectra look quite similar (look identical to me). I guess the authors may also consider citing this paper here because of the great relevance in the frequency extraction method.

Reviewer #2 (Remarks to the Author):

The paper addresses the challenge of achieving high precision in quantum simulations using analog quantum simulators based on superconducting qubits.

In this work, the authors develop a specialised method to accurately determine the full Hamiltonian of a large-scale bosonic system, implemented on superconducting quantum processors. They focus on identifying the non-interacting part of potentially interacting systems and successfully estimate the corresponding Hamiltonian parameters and SPAM errors for all individual components of the superconducting chip, for up to 14-mode Hamiltonians across a wide parameter range.

For robust frequency extraction, they introduce a novel algorithm called tensorESPRIT, combining techniques from super-resolving Fourier analysis and tensor networks. To reconstruct eigenspaces, they employ constrained manifold optimization over the orthogonal group. Importantly, their method explicitly accounts for all structural constraints, enabling the differentiation and acquisition of tomographic information about state-preparation and measurement errors. This information is crucial for making the identification and verification of dynamics experimentally feasible, particularly in quench-based experiments.

The authors support their algorithmic development with numerical simulations to assess various noise effects and benchmark against alternative approaches. They demonstrate the scalability of their method to larger system sizes beyond current experimental capabilities.

I find the paper well written and interesting to read. However, I am concerned that most of the results are empirical and there are no analytic convergence rates of the estimated coefficients to the true ones, at least in the main text. The analysis is done in the supplementary, but mostly covers the algorithms. The results seem really architecture-dependent, and it is not clear how robust they are in reality since the model introduced in the paper is very restrictive. I think the paper is suitable for the Nature communication after addressing these questions a bit more in the main text.

Some of my comments:

1. You impose the structural constraints on the Hamiltonian model: "the Hamiltonian has a spectrum such that the time-series data has a time-independent, sparse frequency

spectrum with exactly
 $\$N\$$ contributions. The Fourier coefficients of the
data have an explicit form...the Hamiltonian
parameter matrix is real and has an a priori known sparse support
due to the experimental connectivity constraints."

I am confused here. Is it your strict demand on the class of Hamiltonians? You tell yourself that these constraints are not respected by various sources of noise. So how can you guarantee that these demands will be fulfilled in practice? Or do you identify these imperfections and not use the data from such realisations? What will happen if some of these demands are not fulfilled, and what are the bounds that your method can tolerate?

2. As I understand it, you introduce the noise model (6) and try to claim that it fits the real data pretty accurately. To this end, you estimated matrices $\$S\$$ and $\$M\$$ by empirical method. Can you give more comments on how you construct these matrices and how general such a model is in the main text? With the growth of the system, the scaling of these matrices must also increase, and empirical methods will require more and more experimental data to provide the desired accuracy. Do you have a scale for this? Also, empirical methods rely on prior knowledge of data distribution, which, for example, can be a problem in the presence of noise.

3. item You claim that the tensorESPRIT is a more efficient method of estimating the eigenvalues of the unknown Hamiltonian than the previous methods.

However, what is the error probability of an incorrect eigenvalue estimation? How does it scale with the system size and the number of experimental shots?

4. Your method is constructed specifically for this type of superconducting architecture and heavily relies on the prior knowledge of the system. Can you give some comments on how efficient the method can be when no knowledge about the system is given or it is different from the theoretically expected one?

5. You use the regularisation method to solve the least-squares optimisation problem (9).

However, you do not indicate how you select the regularisation parameter. You say: "To deal with this, we consecutively

ramp up $\$\mu\$$ until the algorithm does not converge anymore in order to find the Hamiltonian that best approximates

the support constraint while being a proper solution of the optimization problem". Are you using some regularisation parameter selection methods? What is the rate of convergence of your regularised solution to the true one? If you are not using any method, but just randomly selecting the parameter, this method can't be considered robust since any change of the system can dramatically change the optimal value of the regularisation parameter, and you need to look for it blindly again every time.

6. Small remark. In "Author Contributions," one of the authors is missing.

Reviewer #3 (Remarks to the Author):

Referee report for "Robustly learning the Hamiltonian dynamics of a superconducting quantum processor"

The authors present a new theoretical approach to Hamiltonian learning and deploy it on superconducting quantum processors (two versions of the Sycamore processor). Their approach has applications specifically within the arena of 'benchmarking analog quantum simulators'. Whereas benchmarking of digital operations on quantum processors is a well-explored and large field, benchmarking of analog simulators is hitherto less explored. Thus, the paper is timely and the combination of experimental and theoretical developments presented is important for the field at large.

I only have a couple of mostly general comments and considerations to this otherwise very nice paper, and I will mostly be focusing on the experimental details of this work. If the authors address the comments below in an appropriate form, then I am very happy to recommend for publication.

Comments below:

General comment: this paper focuses on characterizing the non-interacting part of the Hamiltonian. Can the authors comment on how one would reasonably estimate the errors

that arise from have non-interacting terms? A typical example for superconducting qubits would be always-on ZZ coupling. If I'm in a situation where I don't know a priori know the strength of this coupling, does the scheme break down? Should I imagine having to bound all the interaction terms before I can run this protocol without fear of uncontrolled errors? Adding a clear discussion of this point will help appreciate the contribution of the paper in a more clear light and highly when this protocol shines (and perhaps when it doesn't).

General comment: a general question on non-unitary dynamics/non-markovian dynamics: as far as I can tell, this method only focuses on the hamiltonian aspects of the evolution, not the Lindbladian dynamics. As the authors mention in the introduction there's been work on Lindblad dynamics as well, but I think it's important to understand what assumptions are made (explicitly and implicitly) for this approach to be used extensively. Some questions:

- How does the authors model fare in the presence of dephasing/amplitude damping on the scale of the experiment? Here it seems the experimental time is on the scale $\sim 200-500$ ns [Fig.1b and 2a], but it is not clear from the manuscript what the T_1 and T_2 times of the qubits in question are. I realize the Sycamore processor has been characterized extensively elsewhere, but it would be very helpful to give coherence details (e.g. in Methods, A. Experimental details). Probably 200 ns / T_2 is some fairly small number, but how does that number play into the limitation of the proposed model?
- There are single-qubit gates used to prepare the relevant input states. What happens if those gates are not 'perfect'? How does this limit the protocol? Stating the fidelity [using CRB, XEB or purity RB or whatever you like] of the single qubit gates is also helpful (e.g. in the methods section).
- Finally, what happens if the noise is non-Markovian? The model assumes \mathcal{H} is time-independent. How much does this break down if there's some non-Markovian effects? Does it all go out the window? Or can we bound some effects?

General comment: there is some sophisticated numerics going on when analysing this data. Could the authors (in an appropriate place, perhaps in supp. matt.) discuss the wall-clock time associated with running this analysis? Of course, factor out data acquisition times etc, but just giving an estimate of how long tensorESPRIT takes to run on a typical workstation,

for e.g. the 5-mode hamiltonian data analyzed in Fig. 2, would be useful for other practitioners in the field.

General comment: the authors find that pulse ramping plays an important role. I'm not surprised as these are typically complex to work with, for flux tunable couplers/qubits. In Supp. Matt. Sec. VII A the authors discuss how to validate their model (its excellent reading!), but it would be great if the authors could supply details on the waveforms used to implement these pulses. Have they been predistorted (I suspect so?), what are the typical timescales of the distortions found? If not pre-distorted, do the authors have a feeling for what predistortion would do to alleviate this issue?

Specific comment: in the abstract, 4th sentence 'we achieve the required levels of precision' -- from the sentences prior it is not clear what 'required level' is or what it refers to.

Required to do what?

Specific comment: in the abstract, 4th sentence the authors refer to 'the model structure', but 'the model' has not been explicitly mentioned previously in the abstract. This makes for slightly clunky reading, what is 'the model'?

Specific comment: in the abstract, final sentence: I think 'fully characterized' is a little strong, since as far as I understand you haven't characterized non-unitary dynamics nor non-markovian effects? Perhaps I'm missing something?

Response to Reviewer 1

In the submitted manuscript, the authors presented a way to robustly learn the free Hamiltonian parameters of a superconducting qubit analogue quantum simulator by measuring the single-mode canonical coordinates. The authors developed new estimation methods that are robust against state preparation and measurement (SPAM) errors. The authors demonstrated this algorithm on two Sycamore processors and achieved high estimation accuracy. Experimental Hamiltonian learning is a topic of increasing interest and involves efforts from both experimental and theoretical perspectives. The methods presented in the manuscript look promising and could stimulate follow-up research in Hamiltonian learning. Therefore, I would recommend the publication of this manuscript in Nature Communications after revisions to the methods.

We would like to start by thanking the reviewer for the thoughtful and helpful report. We are also extremely happy to read the positive assessment and the recommendation of the “the publication of this manuscript in Nature Communications after revisions to the methods”. We have taken these suggestions very seriously and explain in this reply letter what we have done to accommodate the suggestions.

Before its acceptance, since the main contribution of this work is an experimental proposal for free Hamiltonian learning, I would suggest the authors address my following concerns regarding the methods to strengthen the presentation and significance of this work.

In this work, the eigenstate of H lies in the single-excitation subspace, which is a strong constraint for the free Hamiltonian. As a result, the dimension of the system is just N instead of $\text{Poly}(N)$ in the case of more general free Hamiltonians. This consideration greatly decreases the difficulty of learning: in the frequency extractions, we only need to deal with N different frequencies, as indicated by Eq 7. Because of these strict constraints, I doubt whether the methods could be useful in relatively complicated cases. I'll elaborate on my concerns below.

Our results for the first time demonstrate how an intermediate-scale Hamiltonian can be learned robustly and accurately in the presence of noise and device imperfections. We indeed heavily make use of ‘constraints’ on the Hamiltonian (non-interacting, particle-number preserving, sparse) in our work and make use of the available experimental control in order to simplify the analysis as much as possible. We conceive of this as a feature rather than a bug because *enforcing* constraints that we know to be true of the actually implemented Hamiltonian yields a great deal of robustness to errors.

We agree that generalizing our method beyond the setting we consider is not immediately obvious, and yet, as we elaborate in detail in response to the detailed comments below, nothing hinders us in principle from generalizing our approach to, say, interacting bosonic dynamics or using ‘training’ data from higher-particle number input states. We expect that the efficiency of the method stays in principle comparable at the cost of higher-order polynomial optimization in classical post-processing. The latter might of course present further practical numerical challenges and working out the details of these generalizations would go beyond the scope of this work.

Even in the seemingly simple setting of non-interacting dynamics, our work demonstrates that a great deal of method development was necessary in order to make Hamiltonian learning feasible. Let us explain our perspective in detail by addressing your questions point-by-point.

1. When considering free Hamiltonians with many particles, more eigenfrequencies will emerge, and more importantly, the eigenspaces $\langle v_k | \langle v_l |$ will become more complicated. I am curious if this method can deal with this case.

There are two ways we can understand your question.

First, we can interpret it as asking about *particle-number preserving* (PNP) quadratic Hamiltonians evolving initial states in higher-particle sectors. Second, we can interpret it as asking about general, *non-particle number* (NPNP) preserving quadratic (free) Hamiltonians. Let us provide an answer on both interpretations of the question.

Regarding higher particle numbers in input states evolved under PNP free Hamiltonians, it is an important question regarding the learnability and predictive power of our method. The answer is simple if we assume that our device consistently implements the same dynamics of a free Hamiltonian independent of the input state. In order to learn a free PNP Hamiltonian, we use the fact that such a Hamiltonian is already determined by the evolution on zero- and one-particle states, which our device can reliably prepare. Our method has predictive power for evolution under the same Hamiltonian with higher particle numbers in the input, since the dynamics in the higher particle sectors are just given by the symmetrized tensor power of the single-particle dynamics. In particular, the eigenvectors are completely determined by the eigenvectors of the Hamiltonian in the single-particle subspace.

Using this representation theoretic structure, we could in principle use higher-particle number input states in order to learn a free Hamiltonian. Our algorithm also generalizes to the setting, e.g. enforcing additionally a tensor product structure in the diagonalization step of tensorESPRIT. For fixed particle numbers, the number of data points increases and the post-processing will be more difficult (higher-order polynomial optimization); but strictly speaking still computationally efficient (up-to potential initialization problems). We here in a sense just make the simplest choice.

Of course the assumption might actually be violated in practice by collective effects affecting the implementation of the free Hamiltonian depending on the initial state. Here a separate protocol only verifying the assumption rather than learning an input-state dependent Hamiltonian implementation might be more relevant.

Second, regarding NPNP free evolution, let us first remark that the model which naturally governs the dynamics of the Sycamore chip is the PNP model, and hence we focus our study on this model. As we demonstrate, already this model presents formidable signal-processing challenges, which we solve.

Having said that, our method can be adapted even for NPNP free Hamiltonians. In this case we need to measure two distinct $N \times N$ matrix time-series.

- $y_{\{ij\}[l]} = \langle a_j(t_l) \rangle_{\psi_i}$, where $|\psi_i\rangle = \frac{1}{\sqrt{2}} (1 + a_i^\dagger) |vac\rangle$, created from the zero state via an $\pi/2$ X pulse
- $z_{\{ij\}[l]} = \langle a_j(t_l) \rangle_{\phi_i}$, where $|\phi_i\rangle = \frac{1}{\sqrt{2}} (1 + i * a_i^\dagger) |vac\rangle$, created from the zero state via an $\pi/2$ Y pulse

These two $N \times N$ matrix time-series can be combined into a single $2N \times 2N$ matrix time-series, which will be (ignoring noise) given by

$$x_{\{ij\}}[l] = (e^{-i t_l G})_{\{ij\}},$$

where G is the (not necessarily diagonalisable) matrix defined in Eq. (4) of Vincent P Flynn et al 2020 New J. Phys. 22 083004. From the matrix time-series x , tensorESPRIT is able to recover the spectrum of G , even in the case when G is not diagonalizable. However, conjugate gradient descent will only be applicable in the cases when G is Hermitian. In the case when G is diagonalisable, but not unitarily diagonalisable, the group structure cannot be used anymore and we would have to apply methods for optimization over $GL(2N)$, which are generally less reliable.

This generalization beyond particle-number preserving dynamics is an interesting question. Yet, since it extends beyond the focus of our manuscript and because we do not have controlled experimental data on this model, we have decided to only mention the idea sketched above in the outlook (page 7).

2. In continuation to the above point, the authors proposed to cast the eigenspace identification problem into a non-convex optimisation problem in Eq 9. Since $|v_k\rangle$ contains exponentially many elements, from my naïve understanding, this optimisation over $|v_k\rangle$ in Eq 9 is exponentially costly (though it might be simple for the case studied here). The authors may have mentioned the efficiency of the method, but this does not seem clear to me (especially when the target problem is not that simple, see point 3 as well). Could the authors comment on why the identification is efficient and what is the dependence on the dimension of the Hamiltonian?

As introduced in the manuscript, the vectors $|v_k\rangle$ are eigenvectors of the free Hamiltonian *parameters* (the $N \times N$ matrix h), not the actual free part of the Hamiltonian in the Fock space. Therefore, they are N element real vectors, where N is the number of qubits, rendering the optimization efficient in the system size (involving only matrix arithmetics with dimensions at most quadratic in N). As mentioned above, we can in principle generalize our method to input states with higher particle number using the tensor-product structure of the representations of free dynamics. The eigenvector of the Hamiltonian in the Fock sector with m particles is described by the symmetrized tensor-power of $|v_k\rangle$. Therefore, in such a generalization mainly the polynomial degree of the objective function for the eigenvectors will increase.

3. I believe the above concern primarily stems from my limited understanding of the method in the context of a more general free-Hamiltonian case. I was wondering if learning the parameters requires some prior knowledge about the Hamiltonian parameters. For example, if the form of the Hamiltonian is unknown, could we still learn it? I suppose the answer is no. Even though the Hamiltonian is sparse (since the bosonic Hamiltonian is unbounded and may not be local in the qubit representations), this identification may still be hard. There are not too many discussions on it. In addition, after getting the frequency λ_k , how to learn the parameters (given that v_k is unknown and/or the form of the Hamiltonian is unknown)? To answer my question, the authors do not need to present a method, but I just wanted to confirm what kind of problems can be done using the proposed method.

The assumption that we have a PNP free Hamiltonian just implies that the Hamiltonian is parameterized by an $N \times N$ matrix, h in Eq. (1) (for the NPNP case, a $2N \times 2N$, for a 2-body interacting Hamiltonian a $N^2 \times N^2$ matrix). The learning problem thus reduces to learning these coefficient matrices, which we do for the PNP free case using simple input states and measurements. In our method, we do not require a locality assumption, but can promote locality using regularization techniques.

In short: Yes, our identification method will work for arbitrary free PNP Hamiltonians.

4. The essence of the frequency extraction, or more generally, the spectroscopic methods, is that we are able to observe the dynamics by measurement. In this context, the authors choose to create a single-quasiparticle excitation and measure on the x - p basis (local Pauli basis). However, for a relatively general Hamiltonian (even when it is noninteracting), it is in principle hard to guarantee that the signal associated with the initial state and the observable is non-zero (we could call it 'spectral weight', which is the initial state and the observable dependent). Could the authors comment on how to prepare the initial state and how to choose the observable (for either a simple or a general case)? Again, just some comments are expected.

Our data comprises the entire single-particle subspace and tensorESPRIT is designed to make use of all the data. We are guaranteed not to lose any signal by the unitarity of the time evolution in this subspace (up to dissipative effects). For a single initial state and observable, i.e. one time series of our data set, we can not guarantee a non-vanishing signal. Thus, the referee is correct that a direct spectroscopy of the individual time-traces does not necessarily work.

5. The authors claimed that this TensorESPRIT can learn degenerate spectra. However, I did not find too much evidence in either Method or SI. I am a little confused about what the improvement over the existing approaches is.

We thank the reviewer for pointing out this omission. The evidence supporting this claim is given in Fig. 2 of the SM. E.g. in (a), we start with a matrix h with equally separated eigenvalues and move one of these eigenvalues closer towards a neighboring eigenvalue until degeneracy. We show the performance of ESPRIT and tensorESPRIT in recovering its spectrum as a function of the distance of the eigenvalues. This figure demonstrates that the performance of tensorESPRIT is the same, irrespective of the eigenvalue distance. The subsequent panel demonstrates that this indeed yields improved scalability over ESPRIT. We have added a pointer to the evidence in the Methods section when the claim is presented.

6. In the discussion, the authors claimed, 'Generalising our two-step approach developed here, we expect a polynomial scaling with the dimension of the diagnosed particle sector and therefore remain efficient for diagnosing two-, three- and four-body interactions'. The focus of this paper is the identification of the non-interacting part. The generalisation is not that straightforward as the authors claimed.

Indeed, the generalization is not immediate. Let us elaborate our train of thought here: Changing the initial states to superpositions of two-excitation states and the vacuum state, and changing the observable to quadratic terms in the lowering operators, we obtain data of a similar form to Eq.

(3), only inflated to an $N^2 \times N^2$ matrix time-series (and similarly for higher order interactions). We can run the two step recovery procedure on this data with minor changes, allowing us to obtain the interactions in principle. Since the structure of the problem is slightly altered and we do not have theoretical guarantees for the identification method, in the absence of a detailed numerical study we cannot give a precise number of shots and length of the time-series necessary to get an estimate with a given level of precision. However, the analysis of the method performed in the SM makes us confident that the sample complexity will remain polynomial in the number of qubits. This is why we expect that our method remains “efficient for diagnosing two-, three- and four-body interactions”. It is nonetheless possible that polynomial degree and constants could make the method impractical, requiring unrealistic amounts of resources to get a reasonable precision.

7. A major result of this work is the robustness against SPAM errors, but I did not fully understand this point. It looks unclear how to avoid ‘the noise that affected $y[0]$ is now present in every entry of the new data series’. Following Eq 11, the authors said that ‘We can remedy this effect by using several time points $y[0]$ for the inversion. To this end, we compute the concatenation of data series for different choices of l_0 ’. The authors may answer this in SI, but there is no explanation about the intuitions behind it at least in Methods. Could the authors comment on whether the case is much easier with SPOM error only.

When we say that “the noise that affected $y[0]$ is now present in every entry” we mean the shot noise coming from estimating the expectation value using a finite number of samples, not the SPAM error. Therefore, we can remedy this effect by concatenating time-series multiplied by the inversion of various different choices of $y[0]$, as in Eq. (11). We have clarified our explanation of this approach in the Methods section.

This procedure removes the dependence on either S or M , therefore in the case of SPOM errors only, the time series obtained is given by Eq. (54) of the SM. Using the time series Eq. (54) leads to no systematic error in the estimate of h . If full SPAM errors are present, as the referee points out, Eq. (11) contains the error due to the final ramp that we cannot remove, as discussed in Section IV.B of the SM. Therefore, in the presence of SPAM our method has a systematic error due to the final ramping phase. However, since we impose a lot of structure in our data processing, we achieve partial robustness against non-trivial M , especially for a realistic M . This is demonstrated, e.g., in Fig. 7, where we numerically estimate the size of the systematic error for realistic ramping. For a theoretical analysis of the systematic error, see Section IV.B of the SM, and for its experimental investigation Section VII.A of the SM.

8. I felt confused about why D_M is nearly orthogonal. When I looked at SI, I found that the authors referred to Method C: ‘we expect the final ramp M to be nearly diagonal’, but in Method C it said this will be discussed in detail in SI. I suppose this is a very important finding in this work, but it is not that obvious to me.

We provide the argument in the paragraph “Imbalance between initial and final ramping phase” of Method C and have now made the reference to this paragraph more explicit in the supplemental material. The reason is fundamentally the orders of magnitude difference between the ramping distance of the qubits and the couplers when implementing the Hamiltonians used in this work. While the qubits need to travel a distance on the order of 100 MHz, the couplers need

to travel distances on the order of 10 MHz. Therefore, the coupler ramping is rapid, compared to the qubit ramping, leaving us with a nearly diagonal final ramp. This claim is also supported experimentally in Fig. 6.

9. Could the authors explain why the deviation of the target Hamiltonian (red line) is much larger than that of the identified Hamiltonian (yellow line) in Figure 2. From my understanding, there should be no deviation of the target Hamiltonian in the noiseless case, but I may get wrong on this point.

This figure shows the deviation of the *experimental* data from the data simulated using the two models – the target Hamiltonian with a best fit initial ramp $\{h_0, S_0, 1\}$ and the model recovered by our procedure $\{\hat{h}, \hat{S}, \hat{D}_M\}$. This plot shows that the model recovered using our method can better explain the experimentally observed data and therefore the deviation between \hat{h} and h_0 points to a genuine inaccuracy in the implementation of the target Hamiltonian on the Google Sycamore chip. The remaining deviation has two sources:

1. the statistical error in the measured time series due to final number of shots used to obtain the data,
2. deviations of the actual dynamics of the chip from the on site interacting Bose-Hubbard.

10. I felt confused about the simulations in Figure 4. The authors said that ‘we repeat the analysis reported in Fig. 3 for 5-qubit dynamics on different subsets of qubits’. Does this mean the authors only estimated the Hamiltonians on at most 5 qubits (instead of 27 qubits)? If so, does it make sense to show the deviation of the Hamiltonian and initial maps for 27 qubits in Figure 3.

Yes, in this experiment, we identified the Hamiltonian on various overlapping sets of 5 qubits. Every qubit and coupler thus participates in Hamiltonian time evolution coupled with different neighbors. The idea of the plot is now the following: From the recoveries of each 5-qubit Hamiltonian we can estimate the accuracy of a single component of the device (a coupler or a qubit) when participating in 5-qubit evolution. Our plot shows the average accuracy of each component for the case of 5-qubit evolution.

We here choose 5 qubits to illustrate a simple tradeoff: this choice is more data-efficient than global time-evolution, but already captures the effect of higher-order correlated errors than just 2-qubit evolution, which would be sufficient in the absence of such errors.

11. In Methods D the authors tested the algorithm for larger systems. I was wondering why the error becomes smaller with increasing system size in Figure 7. Does this align with our expectations?

Yes, this aligns with the expectations derived for a simplified diagonal model of the final ramp, see Eq. (66) in the SI. The systematic errors due to ramps decrease with the system size.

12. I am curious about how the authors run the noisy simulation up to 100 qubits. It looks like this process with random unitaries and SPAM errors is not classically stimulable.

Since we are working in the single excitation sector of the Hilbert space, which has dimension linear in the number of qubits, the simulation of the time evolution is not resource-intensive. In fact, the time series data is given by Eq. (6) (note that this is the matrix exponential of the coefficient matrix h , which is an $N \times N$ matrix, rather than the Hamiltonian itself). In the simulations, we add shot noise to the data and apply realistic initial and final ramps according to the model from Section VII.A of the SM.

13. I am not an expert in commenting on the experimental details, but I am curious about why the energy scale is quite different from the author Roushan's previous work [Science. 2017 Dec 1;358(6367):1175-1179. doi: 10.1126/science.aao1401.], which I believe is quite relevant to this manuscript. In figure 2 of the Science paper, the energy scale is about one order large than that in the submitted manuscript, while the spectra look quite similar (look identical to me). I guess the authors may also consider citing this paper here because of the great relevance in the frequency extraction method.

In both cases, we are using the one-dimensional Harper model. In the 2017 science paper, $\Delta = 50$ MHz and coupling $J = 50$ MHz, and $NQ=9$. In the current manuscript, $\Delta = 20$ MHz and coupling $J = 20$ MHz, and $NQ=6$. These differences in parameters explain the observed difference in eigen-energy scales. We are surprised ourselves that the reference was not in the manuscript and would like to thank the referee for pointing this out. We have added the reference as Ref. [60]. Again, we thank the reviewer for the extraordinarily careful and helpful report. Having accommodated all detailed comments, we hope that our work is now suitable for publication.

Response to Reviewer 2

The paper addresses the challenge of achieving high precision in quantum simulations using analog quantum simulators based on superconducting qubits. In this work, the authors develop a specialised method to accurately determine the full Hamiltonian of a large-scale bosonic system, implemented on superconducting quantum processors. They focus on identifying the non-interacting part of potentially interacting systems and successfully estimate the corresponding Hamiltonian parameters and SPAM errors for all individual components of the superconducting chip, for up to 14-mode Hamiltonians across a wide parameter range.

For robust frequency extraction, they introduce a novel algorithm called tensorESPRIT, combining techniques from super-resolving Fourier analysis and tensor networks. To reconstruct eigenspaces, they employ constrained manifold optimization over the orthogonal group. Importantly, their method explicitly accounts for all structural constraints, enabling the differentiation and acquisition of tomographic information about state-preparation and measurement errors. This information is crucial for making the identification and verification of dynamics experimentally feasible, particularly in quench-based experiments.

The authors support their algorithmic development with numerical simulations to assess various noise effects and benchmark against alternative approaches. They demonstrate the scalability of their method to larger system sizes beyond current experimental capabilities.

I find the paper well written and interesting to read. (...) I think the paper is suitable for the Nature Communication after addressing these questions a bit more in the main text. However, I am concerned that most of the results are empirical and there are no analytic convergence rates of the estimated coefficients to the true ones, at least in the main text. The analysis is done in the supplementary, but mostly covers the algorithms. The results seem really architecture-dependent, and it is not clear how robust they are in reality since the model introduced in the paper is very restrictive.

Thank you very much for the thoughtful and detailed report. The summary is an accurate summary of what we have done in our work. Also, we are delighted to read that you find “the paper well written and interesting to read” and regard our paper as suitable for publication in Nature Communications.

We have taken your criticism to heart and respond to your comments in detail below. To address these comments, we have clarified the status of our results at several points in the manuscript.

Please allow us to begin by briefly outlining the philosophy of our approach and its relation to other works on the topic, though. Hamiltonian learning has become an important field of research in the past few years, starting around the time that we originally posted our paper. Most of the results in Hamiltonian learning are focused on doing what you suggest—proving recovery guarantees for an abstractly formulated learning task in certain low noise regimes. However, there are only very few examples (all of the ones we are aware of we mention in the introduction of our manuscript) in which Hamiltonian learning has been successfully performed in practice. This is because obtaining robust methods that work using actual data from real experiments is a slightly different and difficult problem in itself.

In the larger body of literature from the past few years, in order to obtain recovery guarantees the focus has been on post-processing algorithms that are more easily amenable to theoretical analysis. Our approach is closer in methods to experimental physics and applied numerics. We start from a very specific learning task and with elaborate signal processing and a manifold of algorithmic tricks, we exploit all available information in order to achieve the highest possible levels of robustness, which—as we demonstrate—are in fact required for an accurate, real experimental realization of Hamiltonian learning. We numerically test and experimentally demonstrate the success of our approach in an interesting instance that already goes beyond a proof-of-concept. The experiment that we report on already provides important diagnostic information to our experimental collaborators that is currently used to improve, e.g., control models and schedules.

As we further highlight in the revised version, our approach is not limited to the specific superconducting qubit platform of the experiment. Although it was developed with this platform in mind, it is applicable to other dynamical analog simulation platforms with a high-degree of local control. Our work illustrates in a case study that making Hamiltonian learning work for a seemingly simple example requires a significant amount of algorithm development and understanding of the experimental platform. We take this as a key takeaway of our work that we believe to be of interest to a broad community. Starting from this solid base, we can then understand how the constraints of different architectures differ and how they can be exploited.

Some of my comments:

1. You impose the structural constraints on the Hamiltonian model: "the Hamiltonian has a spectrum such that the time-series data has a time-independent, sparse frequency spectrum with exactly N contributions. The Fourier coefficients of the data have an explicit form...the Hamiltonian parameter matrix is real and has an a priori known sparse support due to the experimental connectivity constraints." I am confused here. Is it your strict demand on the class of Hamiltonians? You tell yourself that these constraints are not respected by various sources of noise. So how can you guarantee that these demands will be fulfilled in practice? Or do you identify these imperfections and not use the data from such realisations? What will happen if some of these demands are not fulfilled, and what are the bounds that your method can tolerate?

The goal of our approach is to identify the best Hamiltonian approximation to the actually implemented time evolution within a class of Hamiltonians that the simulator attempts to implement with high-precision (non-interacting, particle-number preserving, nearest-neighbor local). Imposing these structural constraints allows us to reduce the complexity of the signal-processing problem and increase the robustness of the recovery to noise. Our analysis and numerical simulations shows that these constraints are not respected by various sources of noise and, therefore, their effect on the recovery error is small so that our model yields good predictive power. So again, our work provides the diagnostic tool for that and those assumptions do not have to be made.

The actual implemented time evolution may involve interactions, it may also involve particle-loss but these are properties that we are not interested in quantifying for the purpose of this work. Eventually, the quality of the recovered Hamiltonian satisfying the constraints that we impose can be read off from the quality of fit to the experimental data, which we show to be small in various

instances, see, e.g., Figs. 2 and 5. The experimental data thus shows that in fact the constraints are approximately satisfied *in reality* by the time evolution of the device. We, thereby, demonstrate that our method is robust against the real experimental imperfections. In the more detailed analysis in the supplemental material, we explicitly account for shot noise and models of the experimentally dominant systematic state preparation and measurement errors. We emphasize that in none of the experimental results shown do we discard any data.

In the case when the implemented time evolution differs significantly from coherent evolution under a Hamiltonian satisfying the constraints we impose, our method will still find a “best fit” candidate from the ansatz class, but the fit of the prediction given the recovered Hamiltonian and SPAM errors to the experimental data will be bad. Therefore, such instances can be detected.

We would like to thank Referee 2 for highlighting a potential confusion about our work and we have included clarification paragraphs regarding this point on pages 2 and 4 in the revised version.

2. As I understand it, you introduce the noise model (6) and try to claim that it fits the real data pretty accurately. To this end, you estimated matrices S and M by empirical method.

This is exactly right; but we do not merely “claim it fits the real data”. Rather, we explicitly show the quality of fit from the prediction of the recovered model of the time evolution to the experimental data.

Can you give more comments on how you construct these matrices and how general such a model is in the main text? With the growth of the system, the scaling of these matrices must also increase, and empirical methods will require more and more experimental data to provide the desired accuracy. Do you have a scale for this? Also, empirical methods rely on prior knowledge of data distribution, which, for example, can be a problem in the presence of noise.

Let us start with recalling the experimental specifics before explaining the generality of the model. For the concrete experiment we perform, we have a good theoretical understanding of the dominant effects modeled by S and M . And we also empirically test these hypotheses with the experiments detailed in Section VII of the SM. In the phase after the state preparation the Hamiltonian parameters are ramped to their target values. Since the evolution is particle-preserving to good approximation these ramp phases are completely described by unitaries S and M in the subspace we measure. Given a schedule for the ramps the unitaries can be computed, e.g., via numerical integration. We also perform the numerical simulations with S and M using this model. However, the formulation of S and M as arbitrary invertible matrices is more general. They can be regarded as the projection of a more general channel to the single-particle subspace providing an effective description in the subspace that our experiment probes. We have extended our discussion when introducing the matrices S and M in the main text.

Our procedure identifies S and M . To this end, we obtain S by linear inversion and a diagonal orthogonal approximation D_M to M by matching the signs of the coupling terms.

As we increase the system size to N modes, S and M will be $N \times N$ matrices like the Hamiltonian parameter matrix itself. The estimation of S and M makes use of the same time trace data as the Hamiltonian learning. The complete data scales quadratically with the system size. Likely, compressed sensing capabilities can be exploited to further subsample the data as numerically illustrated for the eigenvector reconstruction.

Contrary to intuition, perhaps, we in fact expect the systematic error induced by S and M caused by ramping phases to *decrease* with the system size. This is argued in Eq. (66) of the SM and can be seen from the numerics in Fig. 7 of the main text.

3. You claim that the tensorESPRIT is a more efficient method of estimating the eigenvalues of the unknown Hamiltonian than the previous methods. However, what is the error probability of an incorrect eigenvalue estimation? How does it scale with the system size and the number of experimental shots?

It is correct that we do not provide theoretical guarantees on the output of tensorESPRIT in this work. We however extensively benchmark the algorithm numerically with results shown in Fig. 1 and 2 of the SM. Fig. 1 shows the performance of tensorESPRIT for three system sizes as the number of shots increases. In panel (a) of Fig. 2 we demonstrate that tensorESPRIT can recover the spectrum of h even in the case when it is degenerate, which was impossible using previous methods, e.g. ESPRIT. In Fig. 2 panel (b) we show that the performance stays roughly constant with increasing system size. In Fig. 3 panel (c) we demonstrate the robustness against ramping errors. In all these results we find that tensorESPRIT consistently for all trials identifies the eigenspectrum to the precision limited by shot noise, that is, we do not find a failure event. Fig. 1 shows that the l -infinity frequency norm error scales roughly as $1/\sqrt{N}$ with the number of shots N . In the tested regimes the precision stays constant with the system size.

Finding analytical guarantees for super-resolving algorithms, such as ESPRIT, is an active area of research, see, e.g., W. Li, W. Liao and A. Fannjiang 2020 IEEE Transactions on Information Theory 66:7, 4593-4608. These methods are usually sensitive to frequency spacing in a complicated way and finding tight bounds on their performance is highly non-trivial. A similar analysis for the more complicated tensorESPRIT is even more challenging. We have therefore left deriving analytical recovery guarantees for future work.

4. Your method is constructed specifically for this type of superconducting architecture and heavily relies on the prior knowledge of the system. Can you give some comments on how efficient the method can be when no knowledge about the system is given or it is different from the theoretically expected one?

Our method can be used to benchmark the free part of any bosonic particle number preserving Hamiltonian implement on any platform. For the scheme to directly translate we require decent state preparation and measurement of states with single particle excitations. In principle, our method can also be generalized to make use of data of time traces with higher particle number excitations at the price of higher-degree polynomial object functions and the corresponding numerical challenges. It will be accurate in case the errors are dominated by statistical and ramping errors. Otherwise, it will return the best-fit parameters within our model class. But as discussed in more detail in response to your question 1: Any deviations of the experimental data to the model will be detectable in the fit error.

Regarding efficiency, our method will always have the same post-processing scaling. The major issue in case the experimental data is not well modeled by our ansatz might be that the conjugate-gradient optimization does not converge to a good solution. Again, such an instance is detectable, though.

We have also added a corresponding comment to the outlook of the manuscript on page 7.

5. You use the regularisation method to solve the least-squares optimisation problem (9). However, you do not indicate how you select the regularisation parameter. You say: "To deal with this, we consecutively ramp up μ until the algorithm does not converge anymore in order to find the Hamiltonian that best approximates the support constraint while being a proper solution of the optimization problem". Are you using some regularisation parameter selection methods? What is the rate of convergence of your regularised solution to the true one? If you are not using any method, but just randomly selecting the parameter, this method can't be considered robust since any change of the system can dramatically change the optimal value of the regularisation parameter, and you need to look for it blindly again every time.

We thank the referee for this question. Let us first point out that the least-square problem can be solved without regularization. We then simply do not make use of the connectivity constraint of the platform to further improve our estimate. We find the unregularized manifold optimization to be well-conditioned and the conjugate-gradient optimization to converge rapidly independent of the initialization (see, e.g., Fig. 3 of the SM). We suspect that this is caused by the symmetric structure of the orthogonal group together with the simple objective function.

Using regularization has both proven valuable to slightly improve the accuracy of the estimation and the stability against ramping errors and numerically challenging for the problem at hand. Adding regularizing terms seems to make the optimization landscape significantly more complicated. At some instance-dependent threshold of the regularization parameter, the algorithm in fact ceases to converge at all. This is why our approach to selecting the regularization parameter is the search that we describe on p. 7 of the SM:

"We therefore proceed by running the optimization algorithm for increasing, exponentially spaced values of μ until it does not converge. We then perform binary search over μ to find the largest value of μ such that the algorithm converges."

Furthermore, as we discuss in the Methods,

"In order to avoid that we identify a Hamiltonian from a local minimum of the rugged landscape, we only accept Hamiltonians that achieve a total fit of the experimental data within a 5% margin of the fit quality of the unregularized recovery problem, and use the Hamiltonian recovered without the regularization otherwise"

This gives us a systematic method to enforce the locality constraints using regularization whenever the data admits a good description by a locality-constrained Hamiltonian. The structured search approach is "blind" but it is efficient over an exponential parameter range and finds an ϵ -precise

optimal regularization parameter with $\log(1/\epsilon)$ overhead. In a sense we borrow the standard approach barrier methods.

We have clarified in the main text on p. 4 that the use of regularization is optional in the following sentence:

“Additionally, we can incorporate the connectivity constraint on the coefficient matrix h by making use of regularization techniques [61].”

Again, we thank the reviewer for the comments that have helped us prepare a better version of our manuscript. Having accommodated the comments, we hope that our work is now suitable for publication.

Response to Reviewer 3

The authors present a new theoretical approach to Hamiltonian learning and deploy it on superconducting quantum processors (two versions of the Sycamore processor). Their approach has applications specifically within the arena of 'benchmarking analog quantum simulators'. Whereas benchmarking of digital operations on quantum processors is a well-explored and large field, benchmarking of analog simulators is hitherto less explored. Thus, the paper is timely and the combination of experimental and theoretical developments presented is important for the field at large.

I only have a couple of mostly general comments and considerations to this otherwise very nice paper, and I will mostly be focusing on the experimental details of this work. If the authors address the comments below in an appropriate form, then I am very happy to recommend for publication.

We would like to thank the reviewer for the very helpful comments. We are delighted to read that the reviewer thinks this is a "very nice paper" and that they recommend publication, provided we address the comments added. We have taken these comments seriously and discuss here what we have done to accommodate those concerns.

Comments below:

1. General comment: this paper focuses on characterizing the non-interacting part of the Hamiltonian. Can the authors comment on how one would reasonably estimate the errors that arise from have non-interacting terms? A typical example for superconducting qubits would be always-on ZZ coupling. If I'm in a situation where I don't know a priori know the strength of this coupling, does the scheme break down? Should I imagine having to bound all the interaction terms before I can run this protocol without fear of uncontrolled errors? Adding a clear discussion of this point will help appreciate the contribution of the paper in a more clear light and highlight when this protocol shines (and perhaps when it doesn't).

We thank the reviewer for this comment. Our protocol can be run worry-free in case the experiment can guarantee an approximately particle-number preserving dynamics. In this case, the interactions (of the bosons) are simply not visible to our protocol. Your example of ZZ coupling is particle-number preserving and therefore appears in the single-particle dynamics as an effective phase. If one now uses the learned Hamiltonian to predict time-trace data on states with more than one particle, the simulation will of course not predict the effects of interaction terms correctly.

To address these points, we have added discussion in the manuscript (1) when we introduce the constraints imposed on the learned Hamiltonian (page 3), (2) discuss the model of data (page 4), and (3) we have added a new paragraph clarifying the predictive power of our approach (page 5).

2. General comment: a general question on non-unitary dynamics/non-markovian dynamics: as far as I can tell, this method only focuses on the hamiltonian aspects of the evolution, not the Lindbladian dynamics. As the authors mention in the introduction there's been work on Lindblad dynamics as well, but I think it's important to understand what assumptions are made (explicitly and implicitly) for this approach to be used extensively. Some questions:

1. How does the authors model fare in the presence of dephasing/amplitude damping on the scale of the experiment? Here it seems the experimental time is on the scale $\sim 200\text{-}500\text{ns}$ [Fig.1b and 2a], but it is not clear from the manuscript what the T_1 and T_2 times of the qubits in question are. I realize the Sycamore processor has been characterized extensively elsewhere, but it would be very helpful to give coherence details (e.g. in Methods, A. Experimental details). Probably $200\text{ ns} / T_2$ is some fairly small number, but how does that number play into the limitation of the proposed model?

Our method is robust against (small) incoherent Markovian errors; it finds the “best fit” Hamiltonian of the form Eq. (1) that describes the dynamics of the leading eigenvector of the density matrix. We see this very explicitly for the case of particle loss, which simply amounts to a damping of oscillation amplitude in the data. Our qubits are transmons with tunable frequencies and interqubit couplings. The median value of $T_1 = 16.1\ \mu\text{s}$ and the dephasing times T_2 , measured via Ramsey interferometry, have a median value of $5.3\ \mu\text{s}$. The T_2 values after CPMG dynamical decoupling sequences have a median of $17.8\ \mu\text{s}$. For this reason, the corresponding amplitude damping in the time-trace data is about 10%. In this regime recovery error of the Hamiltonian is essentially unaffected.

We have added these values to page 8 of the revised manuscript.

2. There are single-qubit gates used to prepare the relevant input states. What happens if those gates are not 'perfect'? How does this limit the protocol? Stating the fidelity [using CRB, XEB or purity RB or whatever you like] of the single qubit gates is also helpful (e.g. in the methods section).

The imperfections in these gates can be accounted for in the matrices S and M , given that the state preparation and measurements do not leave the zero- and one-particle subspace. We have included a clarification regarding this point on page 5 and we thank Referee 3 for pointing out to us the missing information.

We have also added the single-qubit gate fidelity (given by 99.8% as measured using RB) to the corresponding section in the Methods.

3. Finally, what happens if the noise is non-Markovian? The model assumes ρ is time-independent. How much does this break down if there's some non-Markovian effects? Does it all go out the window? Or can we bound some effects?

In such a case, we find the free part of the “best fit” Hamiltonian of the form Eq. (1) that describes the dynamics of the leading eigenvector of the single-particle density matrix. We can therefore certainly say that if the non-Markovian effects are small, so will the deviation on the recovered Hamiltonian.

3. General comment: there is some sophisticated numerics going on when analysing this data. Could the authors (in an appropriate place, perhaps in supp. matt.) discuss the wall-clock time associated with running this analysis? Of course, factor out data acquisition times etc, but just giving an estimate of how long tensorESPRIT takes to run on a typical

workstation, for e.g. the 5-mode hamiltonian data analyzed in Fig. 2, would be useful for other practitioners in the field.

We provide typical runtimes of tensorESPRIT and ESPRIT on desktop hardware in Table 1 in Section VI.A of the SM and typical runtimes of linInvPP, conjGrad, regConjGrad in Table 2 in Section VI.B of the SM.

4. General comment: the authors find that pulse ramping plays an important role. I'm not surprised as these are typically complex to work with, for flux tunable couplers/qubits. In Supp. Matt. Sec. VII A the authors discuss how to validate their model (its excellent reading!), but it would be great if the authors could supply details on the waveforms used to implement these pulses. Have they been predistorted (I suspect so?), what are the typical timescales of the distortions found? If not pre-distorted, do the authors have a feeling for what predistortion would do to alleviate this issue?

Indeed, the ramping phase turns out to be the major culprit leading to errors in the experiments. The pulses used in the experiment are indeed pre-distorted. Knowing the filters used on the lines allows us to have some knowledge of how pulses would be distorted and we compensate for it. The fine tune calibration of this distortion comes from using the qubits themselves. We send rectangular pulses to the qubits and monitor the frequency change of the qubits. This allows us to know the response of the lines at the qubits (deviation from rectangle) and compensate for distortions. The ramp time can be as fast as 2 to 3 ns and the distortions take the form of overshoot and undershoots with a long response time of ~ 100 ns. After compensating for them, the qubit frequency seems to remain fixed, meaning that the long z-tail has been removed.

Given the complexity of generating the pulses we have for now abstained from attempting to further optimize the pulses. We have however added an explanation of how the pulses are generated to the manuscript.

5. Specific comment: in the abstract, 4th sentence 'we achieve the required levels of precision' -- from the sentences prior it is not clear what 'required level' is or what it refers to. Required to do what?

The levels of precision implied here are those required for a particular analog simulation task. Since we have not looked at a specific task, we have simply changed the sentence to "we achieve high levels of precision".

6. Specific comment: in the abstract, 4th sentence the authors refer to 'the model structure', but 'the model' has not been explicitly mentioned previously in the abstract. This makes for slightly clunky reading, what is 'the model'?

We thank Referee 3 for spotting this clunky formulation and we have replaced "the model" with "a priori knowledge".

7. Specific comment: in the abstract, final sentence: I think 'fully characterized' is a little strong, since as far as I understand you haven't characterized non-unitary dynamics nor non-markovian effects? Perhaps I'm missing something?

That is a good point. We have changed the last sentence to: "Our results constitute an accurate implementation of a dynamical quantum simulation that is characterized using a new diagnostic toolkit for understanding, calibrating, and improving analog quantum processors." Again, we thank the reviewer for the careful comments that have helped us prepare a better version of our manuscript.

List of changes

- Abstract:
 - 'required levels of precision' -> 'high levels of precision'
 - 'the model structure' -> 'a priori knowledge'
 - "Our results constitute a fully characterized, highly accurate implementation of an analog dynamical quantum simulation and introduce a diagnostic toolkit for understanding, calibrating, and improving analog quantum processors." -> "Our results constitute an accurate implementation of a dynamical quantum simulation that is characterized using a new diagnostic toolkit for understanding, calibrating, and improving analog quantum processors."
- Page 2, top of the right column, clarification of the setting
- Page 3, end of the first paragraph of the right column, clarification of the role of the structure in the identification procedure
- Page 4, left column, clarification on the connectivity constraints
- Page 4, right column, comment on state preparation and measurement pulses
- Page 5, added a paragraph on the predictive power of our approach
- Page 7, clarified applicability of the approach
- Page 7, mentioned general quadratic Hamiltonians
- Page 8, first paragraph, added T1 and T2 times
- Page 8, second paragraph, added details on ramping pulses
- Page 8, second paragraph of the right column, pointer to evidence for tensorESPRIT in the SM
- Page 9, right column, clarification regarding the initial ramp removal step
- Page 11, clarification of author contributions

REVIEWERS' COMMENTS

Reviewer #1 (Remarks to the Author):

After reading through the revised manuscript and the reply, the authors have addressed nearly all my previous comments. In particular, the authors discussed the scalability for the theoretical protocol in non-interacting Hamiltonian learning, which I believe is quite impressive and could stimulate follow-up research. As I mentioned in the previous report, I fully acknowledge that enforcing constraints is the key to show robustness to errors, and the importance in learning non-interacting Hamiltonians. I am happy to find that some sections in the main text and SI are improved, highlighting the key results in this work. I would recommend the publication in Nature Communications.

Reviewer #2 (Remarks to the Author):

The authors clarified my questions and added the corresponding explanations to the text. I have no further questions.